# Nucleus accumbens circuit disinhibits lateral hypothalamus glutamatergic neurons contributing to morphine withdrawal memory in male mice

Huan Sheng[1,2,4], Chao Lei[1,4], Yu Yuan[1,4], Yali Fu[1], Dongyang Cui[1], Li Yang[1], Da Shao[1], Zixuan Cao[1], Hao Yang[1], Xinli Guo[1], Chenshan Chu[1], Yaxian Wen[1], Zhangyin Cai [1], Ming Chen [1]✉, Bin Lai [1]✉ & Ping Zheng [1,3]✉

The lateral hypothalamus (LH) is physiologically critical in brain functions. The LH also plays an important role in drug addiction. However, neural circuits underlying LH involvement of drug addiction remain obscure. In the present study, our results showed that in male mice, during context-induced expression of morphine withdrawal memory, LH glutamatergic neurons played an important role; dopamine D1 receptor-expressing medium spiny neurons (D1-MSNs) projecting from the core of nucleus accumbens (NAcC) to the LH were an important upstream circuit to activate LH glutamatergic neurons; D1-MSNs projecting from the NAcC to the LH activated LH glutamatergic neurons through inhibiting LH local gamma-aminobutyric acid (GABA) neurons. These results suggest that disinhibited LH glutamatergic neurons by neural circuits from the NAcC importantly contribute to context-induced the expression of morphine withdrawal memory.

The lateral hypothalamus (LH) emerged as an orchestration in the regulation of sleep–wake states, feeding, stress, reward, and motivated behavior[1,2]. Several studies also revealed an important role of the LH in drug addiction. LH neurons were recruited by stimuli associated with addictive drugs and induced "pathological" drug-seeking behavior[3–5].

There are four main types of neurons in the LH: orexinergic, melanin-concentrating hormone (MCH), GABAergic and glutamatergic neurons[1,6]. Among them, orexinergic neurons in the LH have been shown to play an important role in drug addiction[4,7–9]. Orexinergic neurons have μ-opioid receptors[10,11] and respond to morphine or cocaine administration, withdrawal and related environments[5,11–14]. Knockdown of orexin or using orexin receptor antagonist significantly attenuated morphine or cocaine or related environments-induced reward[13–18] and withdrawal response[10,19]. Increased expression of orexin

in the LH accompanied an increase in ethanol intake[20]. However, the role of glutamatergic neurons of the LH in drug addiction remains poorly documented.

Glutamatergic neuron is one of major cell types in the LH, occupying about 45% of all neurons in the LH[2]. Moreover, almost all orexinergic neurons co-localize with vesicular glutamate transporter 2 (Vglut2), which is a marker of glutamatergic neurons and co-release orexin and glutamate from their axonal terminals[1,2,21]. Co-released orexin may modulate the effect of co-released glutamate via orexin-glutamate interaction by pre- and postsynaptic mechanisms[22,23]. Borgland et al. demonstrated that orexin enhanced the expression of postsynaptic N-methyl-D-aspartate (NMDA) receptors[24]. Intracerebroventricular or intravenous administration of orexin could further promote presynaptic glutamate release[25,26]. Therefore,

[1]State Key Laboratory of Medical Neurobiology, Institutes of Brain Science, MOE Frontier Center for Brain Science, Department of Neurology of Zhongshan Hospital, Fudan University, Shanghai 200032, China. [2]Department of Otorhinolaryngology-Head and Neck Surgery, Zhongshan Hospital, Fudan University, Shanghai 200032, China. [3]Medical College of China Three Gorges University, Yichang 443002, China. [4]These authors contributed equally: Huan Sheng, Chao Lei, Yu Yuan. ✉e-mail: ming_chen@fudan.edu.cn; laibin@fudan.edu.cn; pzheng@shmu.edu.cn

glutamatergic neurons in the LH may play an important role in drug addiction.

Drug addiction has different stages, such as drug reward, drug withdrawal, drug reward memory and drug withdrawal memory[27,28]. The reactivation of drug withdrawal memory by cues or contexts previously associated with withdrawal can drive compulsive drug seeking in abstinent opiate addicts[29]. However, the role of LH glutamatergic neurons in the expression of drug withdrawal memory remains obscure.

It has been known that the LH receives a complex set of cortical and subcortical afferents[17]. Here, we propose that afferents from the nucleus accumbens (NAc) may constitute one upstream circuit that activates LH glutamatergic neurons to participate in context-induced expression of morphine withdrawal memory because the NAc is an important site mediating morphine withdrawal memory[30,31].

The NAc consists of two subregions: the core and shell[32]. In these two subregions, there are two populations of medium spiny projection neurons (MSNs) with a different expression of dopamine receptors: dopamine D1 receptor-expressing MSNs (D1-MSNs) and dopamine D2 receptor-expressing MSNs (D2-MSNs)[33,34]. Previous study showed that the activation of D1-MSNs of NAc shell (NAcSh) enhanced the rewarding effects of cocaine[35], whereas D2-MSNs in the NAcSh were involved in the expression of drug withdrawal memory[36]. Optogenetic long-term depression (LTD)-based in vivo manipulation of the PVT projecting to the D2-MSNs in the NAcSh reduced the expression of morphine withdrawal memory[36]. However, D2-MSNs in the NAcSh have few direct projections to the LH, whereas D1-MSNs in the core of the NAc (NAcC) have denser projections to the LH[37,38]. Therefore, for the activation of LH glutamatergic neurons to participate in the expression of morphine withdrawal memory, the direct projection of D2-MSNs in the NAcSh to the LH may contribute less, whereas D1-MSNs in the NAcC may contribute more.

To test the above hypothesis, firstly, we studied whether context could activate LH glutamatergic neurons in morphine withdrawn mice by examining the influence of context on the co-expression of c-Fos, a marker of neuronal activation, and *Vglut2*, a marker of glutamatergic neurons, in LH neurons using the immunofluorescence staining in combination with fluorescence in situ hybridization, and then studied the role of these neurons in the expression of morphine withdrawal memory by examining the influence of chemogenetic inhibition of LH glutamatergic neurons on conditioned place aversion (CPA). We also used retrograde labeling method to identify NAcC neurons projecting to the LH and examined the co-expression of c-Fos and dopamine D1 receptor in these neurons, and further studied the role of these projection neurons in the expression of morphine withdrawal memory by examining the influence of chemogenetic inhibition of D1-MSNs in the NAcC projecting to the LH on the CPA. At last, by using whole-cell recording and optogenetic method, we studied how D1-MSNs in the NAcC to the LH affected LH glutamatergic neurons to participate in the expression of morphine withdrawal memory.

## Results

### LH glutamatergic neurons play an important role in the expression of morphine withdrawal memory

To study whether context could activate LH glutamatergic neurons in morphine withdrawn mice, we examined the influence of context on the co-expression of c-Fos, a marker of neuronal activation[39], and *Vglut2*, a marker of glutamatergic neurons[40], in LH neurons using the immunofluorescence staining in combination with fluorescence in situ hybridization. Mice were divided into four groups: the saline + saline (SS) group, in which the saline-treated mice were trained to CPA with saline; the saline + naloxone (SN) group, in which the saline-treated mice were trained to CPA with naloxone; the morphine + saline (MS) group, in which the chronic morphine-treated mice were trained to CPA with saline; the morphine + naloxone (MN) group, in which the

chronic morphine-treated mice were trained to CPA with naloxone. Mice in each group experienced a CPA paradigm (Fig. 1a). The results showed that the mice in the MN group exhibited a strong aversion to the withdrawal-paired compartment and thus spent less time in the withdrawal-paired compartment during the post-test than that during the pre-test, resulting in an increase in aversion score (CPA score), whereas mice in other groups did not exhibit a significant aversion to either compartment (two-way ANOVA, drug treatment factor, $F_{(3, 31)} = 21.62$, $p < 0.0001$; test condition factor, $F_{(1, 31)} = 42.26$, $p < 0.0001$; drug treatment x test condition, $F_{(3, 31)} = 24.93$, $p < 0.0001$. Figure 1b). Mice in each group were sacrificed at 90 min after the post-test. c-Fos and *Vglut2* in LH were stained by immunofluorescence and fluorescence in situ hybridization, respectively (Fig. 1c). The average percentage of the c-Fos and *Vglut2* co-labeling neurons relative to *Vglut2* neurons in the LH in the MN group was $18.6 \pm 0.8\%$, which was significantly higher than that in the SS group ($8.5 \pm 1.2\%$), the SN group ($7.7 \pm 0.9\%$), and the MS group ($8.7 \pm 0.8\%$) (One-way ANOVA, $F_{(3, 24)} = 27.32$, $p < 0.0001$. Figure 1d). This result suggests that context can activate LH glutamatergic neurons in morphine withdrawn mice.

To study the role of LH glutamatergic neurons in context-induced expression of morphine withdrawal memory, we examined the influence of in vivo chemogenetic inhibition of LH glutamatergic neurons on context-induced place aversion in morphine withdrawn mice. AAV-DIO-hM4Di-EGFP or AAV-DIO-EGFP was injected into the LH of *Vglut2-cre* mice (Fig. 1f). The mice with the injection of hM4Di were divided into two groups: one group was the saline group, in which the mice received intraperitoneal injection of saline at 40 min before the post-test (hM4Di + saline group); another group was the clozapine-n-oxide (CNO) group, in which the mice received intraperitoneal injection of CNO at 40 min before the post-test (hM4Di + CNO group) to inhibit the activity of LH glutamatergic neurons during the post-test. The mice with the expression of only EGFP without hM4Di were set as the empty vector control group (EGFP + CNO group), in which the mice received intraperitoneal injection of CNO at 40 min before the post-test to exclude the effect of CNO on the CPA (Fig. 1e). The result in Fig. 1g showed that context induced a strong aversion to the morphine withdrawal-paired compartment in the hM4Di + saline group and the EGFP + CNO group, but it did not induce a significant aversion to the morphine withdrawal-paired compartment in the hM4Di + CNO group (two-way ANOVA, drug treatment factor, $F_{(2, 27)} = 33.93$, $p < 0.0001$; test condition factor, $F_{(1, 27)} = 175.8$, $p < 0.0001$; drug treatment x test condition, $F_{(2, 27)} = 26.56$, $p < 0.0001$. Bonferroni's multiple comparisons: the pre-test vs. the post-test in hM4Di + saline ($p < 0.0001$), hM4Di + CNO ($p = 0.1094$) and EGFP + CNO ($p < 0.0001$) groups. The post-test of hM4Di + CNO group vs. hM4Di + saline group: $p < 0.0001$; hM4Di+CNO group vs. EGFP + CNO group: $p < 0.0001$; hM4Di + saline group vs. EGFP + CNO group: $p > 0.9999$. Figure 1g). This result suggests that the activity of LH glutamatergic neurons is required for context-induced expression of morphine withdrawal memory.

### D1-MSNs projecting from the NAcC is an important upstream circuit for the activation of LH glutamatergic neurons in the expression of morphine withdrawal memory

To study whether D1-MSNs or D2-MSNs of the NAcC were upstream circuit of the activation of LH glutamatergic neurons during context-induced expression of morphine withdrawal memory, firstly, we examined the role of D1-MSNs or D2-MSNs of the NAcC in context-induced expression of morphine withdrawal memory and then studied the relationship between them and LH glutamatergic neurons. *D1-cre* or *D2-cre* mice in each group experienced a CPA paradigm (Fig. 2a). To study the role of D1-MSNs in context-induced expression of morphine withdrawal memory, we examined the influence of chemogenetic inhibition of D1-MSNs in the NAcC on context-induced place aversion in morphine withdrawn mice. AAV-DIO-hM4Di-mCherry or AAV-DIO-mCherry was bilaterally injected into the NAcC of *D1-cre* mice (Fig. 2b).

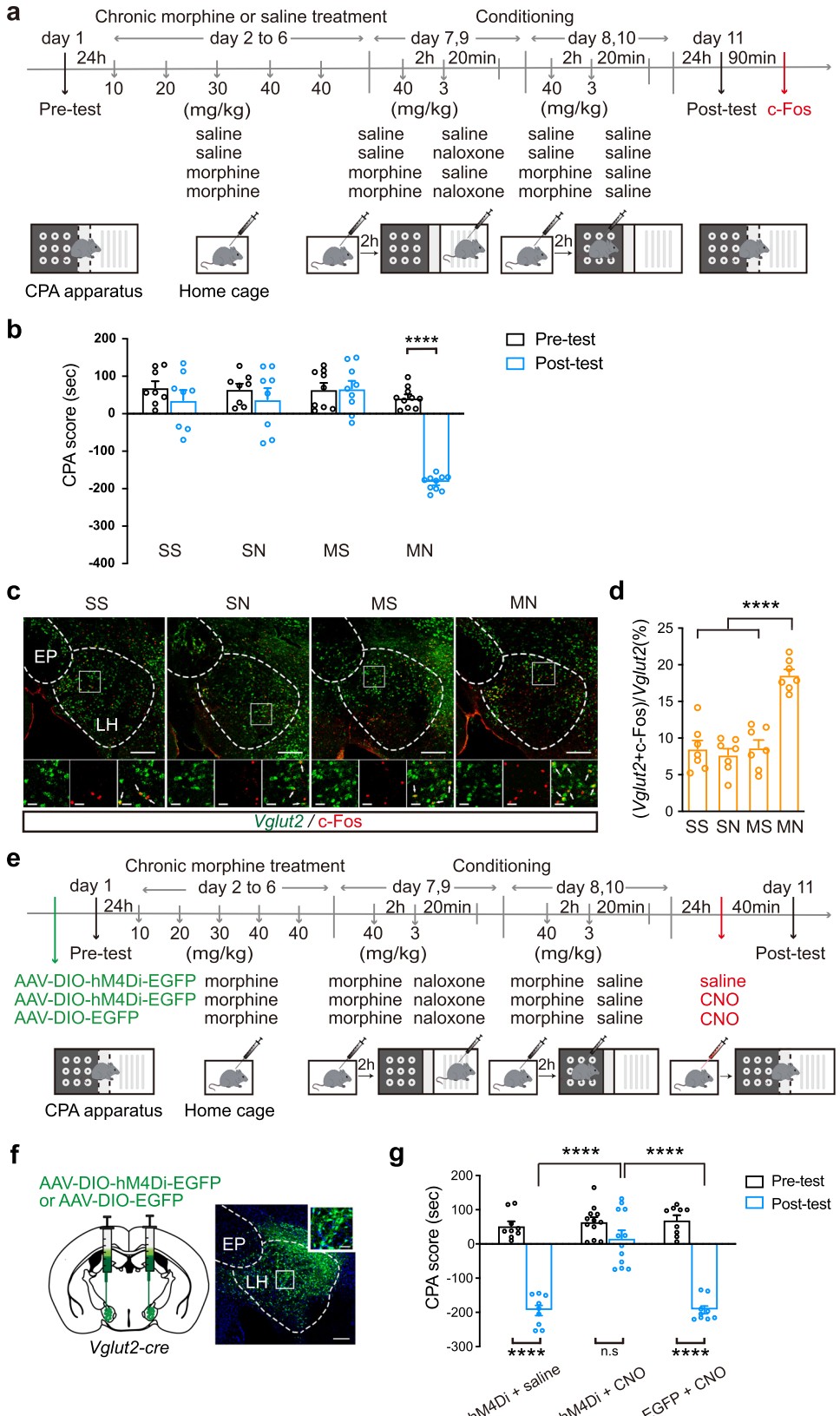

The mice with the injection of hM4Di were divided into two groups: one group was the saline group, in which the mice received intraperitoneal injection of saline at 40 min before the post-test (hM4Di + saline group); another group was the CNO group, in which the mice received intraperitoneal injection of CNO at 40 min before the post-test (hM4Di + CNO group) to inhibit the activity of NAcC D1-MSNs

during the post-test. The mice with the expression of only mCherry without hM4Di were set as the empty vector control group (mCherry + CNO group), in which the mice received intraperitoneal injection of CNO at 40 min before the post-test to exclude the effect of CNO on CPA. The result showed that context induced a strong aversion to the morphine withdrawal-paired compartment in the hM4Di + saline

**Fig. 1 | The role of LH glutamatergic neurons in morphine withdrawal memory expression. a** Experimental timeline. **b** The average CPA scores in SS ($n = 8$), SN ($n = 8$), MS ($n = 9$) and MN ($n = 10$) groups. Two-way ANOVA, drug treatment factor, $F_{(3, 31)} = 21.62$, $p < 0.0001$; test condition factor, $F_{(1, 31)} = 42.26$, $p < 0.0001$; drug treatment x test condition, $F_{(3, 31)} = 24.93$, p < 0.0001. **c** Top: The *Vglut2* and c-Fos co-labeling neurons in LH in the four groups. Magnified image shows the boxed area. Scale bar: 200 μm. Bottom: left, *Vglut2* neurons; middle, c-Fos positive neurons; right, c-Fos and *Vglut2* co-labeling neurons. Scale bar: 30 μm. **d** The average percentage of c-Fos and *Vglut2* co-labeling neurons relative to *Vglut2* neurons in LH in the four groups ($n = 7$ in each group). One-way ANOVA, $F_{(3, 24)} = 27.32$, $p < 0.0001$. **e** The experimental timeline. **f** Left: the diagram of the injection of virus

into the bilateral LH (AAV-DIO-hM4Di-EGFP or AAV-DIO-EGFP) in *Vglut2-cre* mice. Right: the expression of hM4Di-EGFP in the LH *Vglut2* neurons. Scale bars, 200 μm and 50 μm. **g** The average CPA scores in hM4Di + saline group ($n = 9$), hM4Di + CNO group ($n = 12$) and EGFP + CNO group ($n = 9$). Two-way ANOVA, drug treatment factor, $F_{(2, 27)} = 33.93$, $p < 0.0001$; test condition factor, $F_{(1, 27)} = 175.8$, $p < 0.0001$; drug treatment x test condition, $F_{(2, 27)} = 26.56$, $p < 0.0001$. Bonferroni's multiple comparisons: the pre-test vs. the post-test in hM4Di + saline ($p < 0.0001$), hM4Di + CNO ($p = 0.1094$) and EGFP + CNO ($p < 0.0001$) groups. The post-test of hM4Di + CNO group vs. hM4Di + saline group: $p < 0.0001$; hM4Di+CNO group vs. EGFP + CNO group: p < 0.0001; hM4Di + saline group vs. EGFP + CNO group: $p > 0.9999$. Means ± SEMs. ****$p < 0.0001$.

group and the mCherry + CNO group, but it did not induce a significant aversion to the morphine withdrawal-paired compartment in the hM4Di + CNO group (Two-way ANOVA, drug treatment factor, $F_{(2, 17)} = 7.709$, $p = 0.0041$; test condition factor, $F_{(1, 17)} = 50.57$, $p < 0.0001$; drug treatment x test condition, $F_{(2, 17)} = 6.443$, $p = 0.0083$. Bonferroni's multiple comparisons: the pre-test vs. the post-test in hM4Di + saline ($p < 0.0001$), hM4Di + CNO ($p = 0.1768$) and mCherry + CNO ($p = 0.0044$) groups. The post-test of hM4Di + CNO group vs. hM4Di + saline group: p = 0.0001; hM4Di + CNO group vs. EGFP + CNO group: $p = 0.0032$; hM4Di + saline group vs. EGFP + CNO group: $p > 0.9999$. Figure 2c). This result suggests that D1-MSNs of the NAcC participate in context-induced expression of morphine withdrawal memory. However, when we used the same strategy in *D2-cre* mice to study the role of D2-MSNs of the NAcC in the context-induced expression of morphine withdrawal memory, we found that chemogenetic inhibition of D2-MSNs of the NAcC did not influence context-induced expression of morphine withdrawal memory. The average CPA score in the post-test at 24 h after the last context training in hM4Di + CNO group was −148.9 ± 51.8 s, which was not statistically different from that in hM4Di + saline group (−175.2 ± 53.7 s) and the mCherry + CNO group (−171.7 ± 47.0 s) (two-way ANOVA, drug treatment factor, $F_{(2, 14)} = 0.3905$, $p = 0.6839$; test condition factor, $F_{(1, 14)} = 56.07$, $p < 0.0001$; drug treatment × test condition, $F_{(2, 14)} = 0.06511$, $p = 0.9372$. Bonferroni's multiple comparisons: the pre-test vs. the post-test in hM4Di + saline ($p = 0.0017$), hM4Di + CNO ($p = 0.0002$) and mCherry + CNO ($p = 0.0104$) groups. The post-test of hM4Di + CNO group vs. hM4Di + saline group: $p > 0.9999$; hM4Di + CNO group vs. EGFP + CNO group: $p > 0.9999$; hM4Di + saline group vs. EGFP + CNO group: $p > 0.9999$. Figure 2e). This result suggests that D2-MSNs of the NAcC do not participate in context-induced expression of morphine withdrawal memory. Therefore, it is possible that D1-MSNs, rather than D2-MSNs, projecting from the NAcC, is an upstream circuit for the activation of LH glutamatergic neurons during context-induced expression of morphine withdrawal memory. To test this hypothesis, we performed the following experiments.

Firstly, we studied whether context could activate NAcC D1-MSNs projecting to the LH by examining the influence of context on the expression of c-Fos in NAcC D1-MSNs projecting to the LH. Alexa Fluor 647 conjugated retrograde tracer cholera toxin subunit B (CTB647) was injected into the LH to retrograde label NAcC projection neurons to the LH (Fig. 3b). After recovery from the surgery of CTB647 injection, mice were subjected to behavioral training as illustrated in Fig. 3a. The results showed that the mice in the MN group exhibited a strong aversion to the withdrawal-paired compartment, whereas mice in other groups did not exhibit a significant aversion to either compartment (two-way ANOVA, drug treatment factor, $F_{(3, 20)} = 11.26$, $p < 0.0001$; test condition factor, $F_{(1, 20)} = 66.09$, $p < 0.0001$; drug treatment × test condition, $F_{(3, 20)} = 27.74$, $p < 0.0001$. Figure 3c). After behavioral assay, animals were sacrificed and slices containing the NAcC were prepared. The co-expression of c-Fos and D1-MSNs marker *D1* in CTB647 labeling neurons in the NAcC was examined using the immunofluorescence staining in combination with fluorescence in situ hybridization (Fig. 3d). Firstly, we observed whether D1-MSNs were

activated in the NAc during the expression of withdrawal memory. The average percentage of c-Fos and *D1* double labeling neurons relative to *D1* labeling neurons in the NAcC in the MN group was 15.1 ± 0.7%, which was significantly higher than that in the SS group (7.1 ± 0.3%), the SN group (7.0 ± 0.4%), and the MS group (7.5 ± 0.4%) (one-way ANOVA, $F_{(3, 20)} = 75.29$, $p < 0.0001$. Figure 3e). This result suggests that context can activate NAcC D1-MSNs in morphine withdrawn mice. Then we examined whether NAcC D1-MSNs projecting to the LH were activated during the expression of withdrawal memory. The average percentage of c-Fos, *D1* and CTB647 triple labeling neurons relative to *D1* and CTB647 double labeling neurons in the NAcC in the MN group was 30.1 ± 2.5%, which was significantly higher than that in the SS group (15.6 ± 1.5%), the SN group (13.1 ± 1.0%), and the MS group (14.7 ± 1.3%) (One-way ANOVA, $F_{(3, 20)} = 24.19$, $p < 0.0001$. Figure 3f). This result suggests that context can activate NAcC D1-MSNs projecting to the LH in morphine withdrawn mice.

Then, to study the role of NAcC D1-MSNs projecting to the LH in context-induced expression of morphine withdrawal memory, we examined the influence of the inhibition of NAcC D1-MSNs projecting to the LH by using dual-virus intersectional strategy on context-induced place aversion in morphine withdrawn mice. Retro AAVs encoding Cre-dependent Flpo (AAV-Retro-FLEX-Flpo) was bilaterally injected into the LH of *D1-cre* mice and at the same time AAVs encoding Flpo-dependent hM4D(Gi)-EGFP (AAV-fDIO-hM4Di-EGFP) was bilaterally injected into the NAcC, which would result in the expression of hM4Di in NAcC D1-MSNs to the LH upon Flpo excision (Fig. 3h). Mice of control group were injected with AAV-Retro-FLEX-Flpo in the LH and AAV-fDIO-EGFP in the NAcC. The mice with the injection of hM4Di-EGFP were divided into two groups: one group was the saline group, in which the mice received intraperitoneal injection of saline at 40 min before the post-test (hM4Di + saline group); another group was the CNO group, in which the mice received intraperitoneal injection of CNO at 40 min before the post-test (hM4Di + CNO group) to inhibit the activity of NAcC D1-MSNs projecting to the LH during the post-test. The mice with the expression of EGFP were set as the empty vector control group (EGFP + CNO group), in which the mice received intraperitoneal injection of CNO at 40 min before the post-test to exclude the effect of CNO on the CPA. Mice of three groups were subjected to behavioral procedure as shown in the Fig. 3g. The result showed that context induced a strong aversion to the morphine withdrawal-paired compartment in the hM4Di + saline group and the EGFP + CNO group, but it did not induce a significant aversion to the morphine withdrawal-paired compartment in the hM4Di + CNO group (two-way ANOVA, drug treatment factor, $F_{(2,28)} = 32.71$, $p < 0.0001$; test condition factor, $F_{(1,28)} = 201.3$, $p < 0.0001$; drug treatment × test condition, $F_{(2,28)} = 46.93$, $p < 0.0001$. Bonferroni's multiple comparisons: the pre-test vs. the post-test in hM4Di + saline ($p < 0.0001$), hM4Di + CNO ($p > 0.9999$) and EGFP + CNO ($p < 0.0001$) groups. The post-test of hM4Di + CNO group vs. hM4Di + saline group: $p < 0.0001$; hM4Di + CNO group vs. EGFP + CNO group: $p < 0.0001$; hM4Di + saline group vs. EGFP + CNO group: $p = 0.7201$. Figure 3i). This result suggests that NAcC D1-MSNs projecting to the LH are essential in context-induced expression of morphine withdrawal memory.

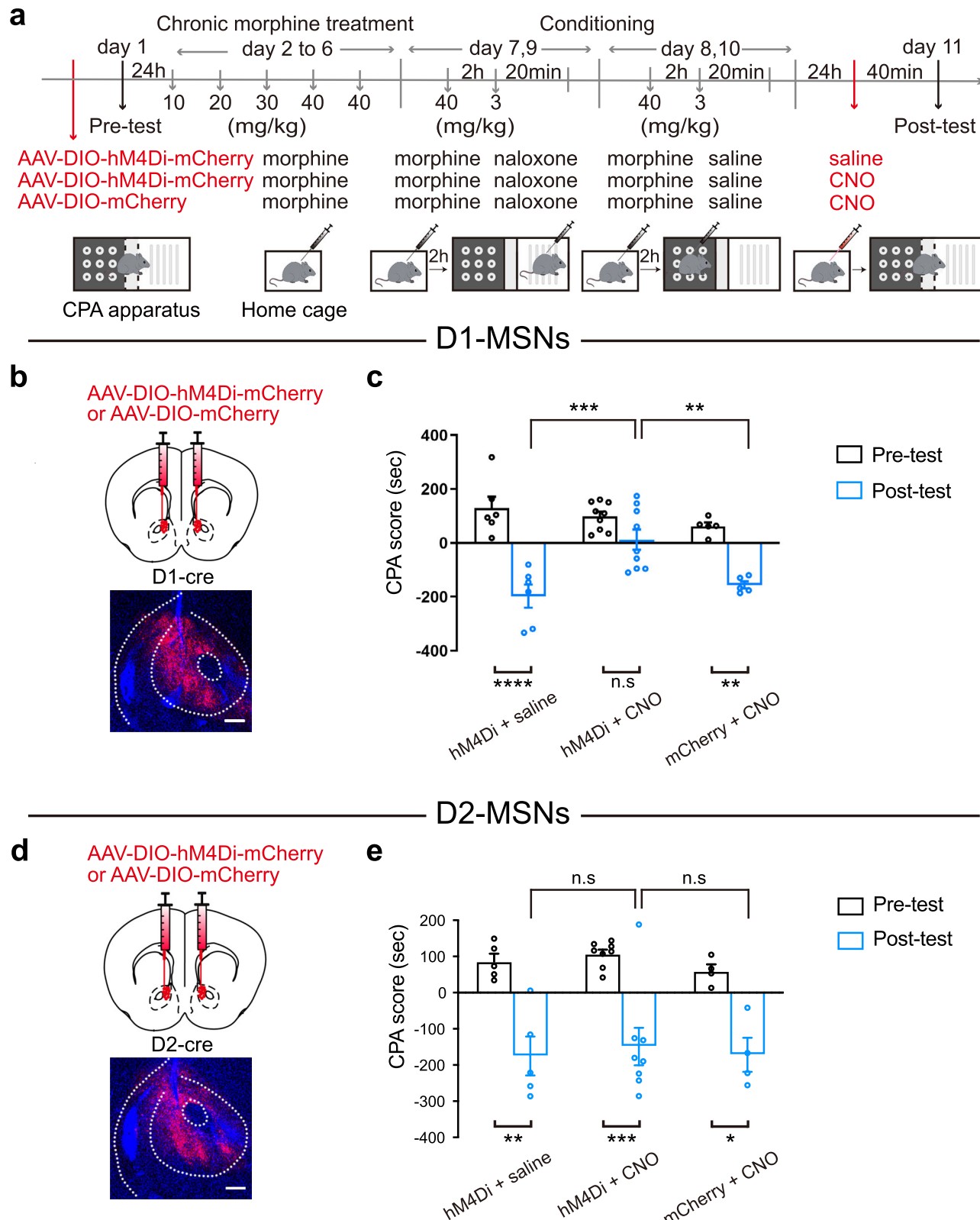

To study whether NAcC D1-MSNs projecting to the LH are upstream circuit of the activation of LH glutamatergic neurons during context-induced expression of morphine withdrawal memory, we quantitatively analyzed the percentage of activated LH glutamatergic neurons without and with the inhibition of NAcC D1-MSNs projecting to LH using chemogenetic method in the morphine withdrawn mice. The mice of each group were sacrificed at 90 min after the post-test

and the percentage of c-Fos and *Vglut2* co-labeling neurons relative to *Vglut2* neurons in the LH was examined by the method of immuno-fluorescence and fluorescence in situ hybridization (Fig. 4a). The result showed that the average percentage of c-Fos and *Vglut2* co-labeling neurons relative to *Vglut2* neurons in the LH in the hM4Di + CNO group was $6.2 \pm 0.6\%$, which was significantly lower than that in the hM4Di + saline group ($15.1 \pm 1.1\%$) and that in the EGFP + CNO group

**Fig. 2 | The influence of chemogenetic inhibition of NAcC D1-MSNs or D2-MSNs on morphine withdrawal memory expression. a** The experimental timeline. **b** Top: the diagram of the injection of virus into the bilateral NAcC (AAV-DIO-hM4Di-mCherry or AAV-DIO-mCherry) in *D1-cre* mice. Bottom: the expression of hM4Di-mCherry in the NAcC. Scale bars, 200 μm. **c** The average CPA score in hM4Di + saline group ($n = 6$), hM4Di + CNO group ($n = 9$) and the mCherry + CNO group ($n = 5$). Two-way ANOVA, drug treatment factor, $F_{(2, 17)} = 7.709$, $p = 0.0041$; test condition factor, $F_{(1, 17)} = 50.57$, $p < 0.0001$; drug treatment x test condition, $F_{(2, 17)} = 6.443$, p = 0.0083. Bonferroni's multiple comparisons: the pre-test vs. the post-test in hM4Di + saline ($p < 0.0001$), hM4Di + CNO ($p = 0.1768$) and mCherry + CNO ($p = 0.0044$) groups. The post-test of hM4Di + CNO group vs. hM4Di + saline group: $p = 0.0001$; hM4Di + CNO group vs. EGFP + CNO group: $p = 0.0032$; hM4Di + saline group vs. EGFP + CNO group: p > 0.9999. **d** Top: the diagram of the injection of virus into the bilateral NAcC (AAV-DIO-hM4Di-mCherry or AAV-DIO-mCherry) in *D2-cre* mice. Bottom: the expression of hM4Di-mCherry (red-colored) in the NAcC. Scale bars, 200 μm. **e** the average CPA scores in hM4Di + saline group ($n = 5$), hM4Di + CNO group ($n = 8$) and the mCherry + CNO group ($n = 4$). Two-way ANOVA, drug treatment factor, $F_{(2, 14)} = 0.3905$, $p = 0.6839$; test condition factor, $F_{(1, 14)} = 56.07$, $p < 0.0001$; drug treatment × test condition, $F_{(2, 14)} = 0.06511$, $p = 0.9372$. Bonferroni's multiple comparisons: the pre-test vs. the post-test in hM4Di + saline ($p = 0.0017$), hM4Di + CNO ($p = 0.0002$) and mCherry + CNO ($p = 0.0104$) groups. The post-test of hM4Di + CNO group vs. hM4Di + saline group: $p > 0.9999$; hM4Di + CNO group vs. EGFP + CNO group: $p > 0.9999$; hM4Di + saline group vs. EGFP + CNO group: $p > 0.9999$. Means ± SEMs. **$p < 0.01$, ***$p < 0.001$,****$p < 0.0001$.

(16.3 ± 1.2%) (One-way ANOVA, $F_{(2, 18)} = 30.07$, $p < 0.0001$. Figure 4b). This result suggests that the activity of NAcC D1-MSNs projecting to the LH is essential to the activation of LH glutamatergic neurons during context-induced expression of morphine withdrawal memory.

To test whether the activation of NAcC D1-MSNs projecting to the LH could increase the excitability of LH glutamatergic neurons, we used whole-cell patch-clamp recording technique in combination with optogenetic activation method to examine the influence of the activation of NAcC D1-MSNs projecting to the LH on firing of action potentials. We crossed *D1-Cre* and *Vglut2-Flpo* mice to label NAcC D1-MSNs and LH glutamatergic neurons in one mouse (*D1-Cre::Vglut2-Flpo* mice). Then, AAVs encoding Cre-dependent channelrhodopsin-2 (AAV-DIO-ChR2-mCherry) was injected into the NAcC to activate D1-MSNs by following photostimulation and AAV-fDIO-mCherry was injected into the LH to label LH glutamatergic neurons in *D1-Cre::Vglut2-Flpo* mice (Fig. 4c, left). Four weeks after virus injection, when D1-MSNs terminals from the NAcC expressed enough ChR2-mCherry and LH glutamatergic neurons expressed mCherry (Fig. 4c, middle), we recorded action potentials of LH glutamatergic neurons using whole-cell patch-clamp recording technique under light-off and blue light-on (470 nm, 2 ms) conditions. Figure 4d shows typical action potential trace in response to 100 pA depolarizing current under light-off condition and blue light-on condition. From these raw traces, we could see that in response to this depolarizing current, neurons under blue light-on condition fired more action potentials than that under light-off condition. The average frequency of action potential firing was 8.75 ± 0.48 Hz under light-off condition and 11.74 ± 0.64 Hz under blue light-on condition, showing a significant increase under blue light-on condition (Paired *t* test, $p < 0.0001$. Figure 4e). We also examined whether longer duration of photostimulation of NAcC D1-MSNs projecting to the LH without simultaneous electrical stimulation was able to increase firing activity of LH glutamatergic neurons. Since LH glutamatergic neurons did not have spontaneous firing under normal conditions, we continuously held neurons at 10−20 pA under a current-clamp mode to induce spontaneous firing (Fig. S1a). Figure S1b, c were the raster plot and peristimulus time histograms (PSTH) of the response of LH glutamatergic neurons to a long blue light stimulation (n = 11 cells). The result showed that the average frequency of spontaneous firing of action potentials under blue light-on was 0.54 ± 0.12 Hz, which was significantly higher than that under light-off (0.24 ± 0.08 Hz) (Paired *t* test, $p = 0.0038$. Figure S1d). These results suggest that the activation of NAcC D1-MSNs projecting to the LH increases the excitability of LH glutamatergic neurons.

It has been known that NAcC D1-MSNs are GABAergic neurons and release GABA after being activated[41,42]. To study the role of GABA in blue light-on-induced increase in the frequency of action potential firing of LH glutamatergic neurons, we examined the influence of bath application of GABA-A receptor antagonist on blue light -on-induced increase in the frequency of action potential firing of LH glutamatergic neurons. Figure 4f shows typical action potential traces in response to 100 pA depolarizing current under light-off, PTX and PTX + blue light-on conditions. From these raw traces, we could see that in response to

the depolarizing current, neurons under PTX condition fired more action potentials than that under light-off condition, but in the presence of PTX, blue light-on could not further increase the frequency of action potential firing of LH glutamatergic neurons. The average frequency of action potential firing was 8.65 ± 1.02 Hz under light-off condition and 11.98 ± 1.30 Hz under PTX condition, showing a significant increase after PTX, but in the presence of PTX, the average frequency of action potential firing was 11.52 ± 1.25 Hz under blue light-on condition, showing no further significant change compared to PTX condition (One-way ANOVA, $F_{(7, 14)} = 20.47$, $p < 0.0001$.Tukey's multiple comparisons, light-off group vs. PTX group: $p < 0.0094$; PTX group vs. PTX + blue light-on group: $p = 0.1953$. Figure 4g). We also repeated this experiment in non-fluorescence labeled LH glutamatergic neurons with the same approaches to those in fluorescence labeled neurons and obtained a similar result (Fig. S1e−h). The average frequency of action potential firing under blue light-on condition was 12.31 ± 0.81 Hz, which was significantly higher than that under light-off condition (9.91 ± 0.63 Hz) (Paired *t* test, $p = 0.0027$. Figure S1f) in non-fluorescence labeled LH glutamatergic neurons. The average frequency of action potential firing in the presence of PTX under light-off condition was 10.00 ± 0.43 Hz, which was higher than that under light-off condition (6.94 ± 1.00 Hz), but there was no significant difference before and after blue light-on in the presence of PTX (9.72 ± 0.67 Hz) in non-fluorescence labeled LH glutamatergic neurons (one-way ANOVA, $F_{(5, 10)} = 5.067$, $p = 0.0142$. Tukey's multiple comparisons: light-off group vs. PTX group: $p = 0.0438$, PTX group vs. PTX + blue light-on group: $p = 0.8547$. Figure S1h). To confirm that recorded non-fluorescence labeled neurons were glutamatergic neurons, we extracted intracellular contents from the recorded neurons of non-fluorescence labeled neurons and performed Single-cell RT-PCR with primers for *Vglut2* (Fig. S1i). These results suggest that the activation of NAcC D1-MSNs projecting to the LH activates LH glutamatergic neurons via GABAergic signaling.

### NAcC D1-MSNs projecting to the LH activate LH glutamatergic neurons through inhibiting LH local GABAergic neurons during morphine withdrawal memory expression

There are two possible approaches through which NAcC D1-MSNs projecting to the LH activate LH glutamatergic neurons: direct innervation and indirect innervation. Since NAcC D1-MSNs projecting to the LH are GABAergic neurons[43], the possibility of the activation of LH glutamatergic neurons by D1-MSNs via direct innervation is very low. Therefore, we propose a hypothesis that context-activated NAcC D1-MSNs GABAergic neurons may produce an inhibitory effect on LH local GABAergic neurons via releasing GABA and then induce a disinhibitory effect of local GABAergic neurons on LH glutamatergic neurons, which lead to an activation of these glutamatergic neurons to participate in the expression of morphine withdrawal memory. To test this hypothesis, we performed the following experiments.

Firstly, we used whole-cell patch-clamp in combination with optogenetic method to examine functional connections between NAcC D1-MSNs and LH neurons. In *D1-Cre::Vglut2-Flpo* mice, AAV-DIO-

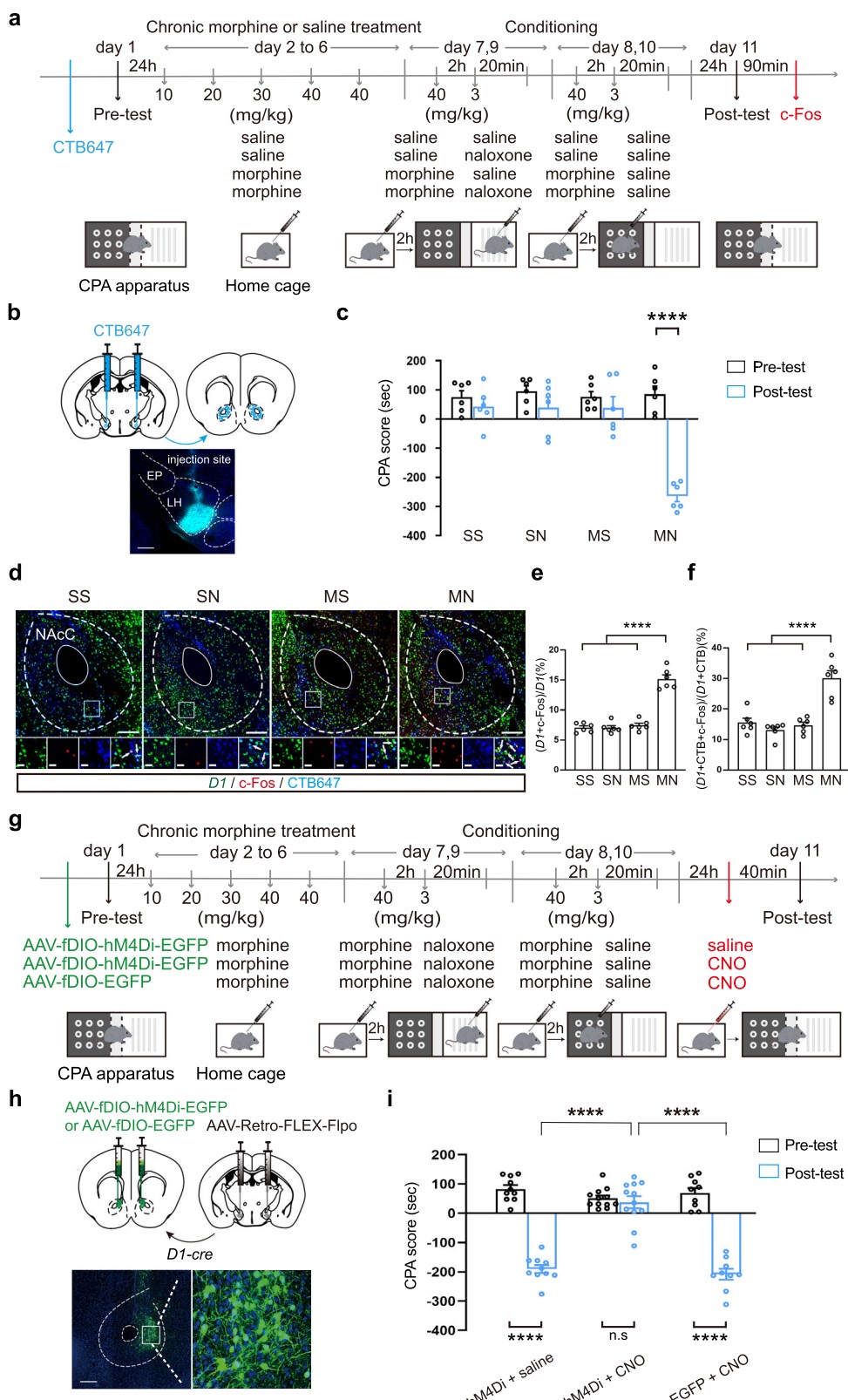

ChR2-mCherry was injected into the NAcC to activate the D1-MSNs by photostimulation, AAV-fDIO-mCherry was injected into the LH to label the LH glutamatergic neurons and AAV-mDlx-EGFP which is controlled by a GABAergic promoter mDlx was injected into LH to label the LH GABAergic neurons[44] (Fig. 5a, top). Four weeks after the injection of virus, whole-cell patch recording was performed in LH glutamatergic neurons or LH GABAergic neurons when D1-MSNs terminals from the NAcC expressed enough ChR2-mCherry (Fig. 5a, middle). The result

showed that single pulses of photostimulation (blue light, 470 nm, 2 ms) could not elicit inhibitory postsynaptic currents (IPSCs) in most LH glutamatergic neurons (Fig. 5b), indicating that NAcC D1-MSNs had few direct innervations on LH glutamatergic neurons. However, single pulses of photostimulation (blue light, 470 nm, 2 ms) could elicit IPSCs in most LH GABAergic neurons and this IPSCs could be blocked by PTX (Fig. 5c). This result suggests that NAcC D1-MSNs mainly innervate LH GABAergic neurons.

**Fig. 3 | The role of D1-MSNs projecting from NAcC to LH in morphine withdrawal memory expression. a** The experimental timeline. **b** Top: the diagram of CTB647 bilateral injection into the LH. Bottom: the injection site of CTB647 in the LH. **c** The average CPA score in SS, SN, MS, MN groups ($n$ = 6 in each group). Two-way ANOVA, drug treatment factor, $F_{(3, 20)}$ = 11.26, $p < 0.0001$; test condition factor, $F_{(1, 20)}$ = 66.09, $p < 0.0001$; drug treatment × test condition, $F_{(3, 20)}$ = 27.74, $p < 0.0001$. **d** Top: the $D1$, c-Fos and CTB647 co-labeling neurons in NAcC in the four groups. Magnified image shows the boxed area. Scale bar: 200 μm. Bottom: first column, $D1$ neurons; second column, c-Fos positive neurons; third column, CTB647 neurons; forth column, $D1$, c-Fos and CTB647 co-labeling neurons. Scale bar: 30 μm. **e** The average percentage of the c-Fos and $D1$ co-labeling neurons relative to the $D1$ neurons in the NAcC in the four groups ($n$ = 6 mice in each group). One-way ANOVA, $F_{(3, 20)}$ = 75.29, $p < 0.0001$. **f** The average percentage of the c-Fos, $D1$ and CTB647 co-labeling neurons relative to the $D1$ and CTB647 co-labeling neurons in the NAcC in the four groups ($n$ = 6 mice in each group). One-way ANOVA, $F_{(3, 20)}$ = 24.19, $p < 0.0001$. **g** The experimental timeline. **h** Left: the diagram of the injection of virus into the bilateral NAcC (AAV-fDIO-hM4Di-EGFP or AAV-fDIO-EGFP) and LH (AAV-Retro-FLEX-Flpo) in $D1$-cre mice; Bottom: the expression of hM4Di-EGFP in the NAcC. Scale bars: 300 μm (left) and 50 μm (right). **i** The average CPA scores in hM4Di + saline group ($n$ = 10), hM4Di + CNO group ($n$ = 12) and the EGFP + CNO group ($n$ = 9). Two-way ANOVA, drug treatment factor, $F_{(2,28)}$ = 32.71, $p < 0.0001$; test condition factor, $F_{(1,28)}$ = 201.3, $p < 0.0001$; drug treatment x test condition, $F_{(2,28)}$ = 46.93, $p < 0.0001$. Bonferroni's multiple comparisons: the pre-test vs. the post-test in hM4Di + saline ($p < 0.0001$), hM4Di + CNO ($p > 0.9999$) and EGFP + CNO ($p < 0.0001$) groups. The post-test of hM4Di + CNO group vs. hM4Di + saline group: $p < 0.0001$; hM4Di + CNO group vs. EGFP + CNO group: $p < 0.0001$; hM4Di + saline group vs. EGFP + CNO group: $p = 0.7201$. Means ± SEMs. ****$p < 0.0001$.

We also examined whether NAcC D1-MSNs terminals released more GABA at presynaptic site of LH GABAergic neurons in CPA model. AAV-DIO-ChR2-mCherry was injected into the NAcC of $D1$-Cre mice to specifically activate the D1-MSNs and AAV-mDlx-EGFP was injected into LH to label LH GABAergic neurons (Fig. 5e, top). Four weeks after the injection of virus, when D1-MSNs terminals from the NAcC expressed enough ChR2-mCherry and LH GABAergic neurons expressed EGFP (Fig. 5e, middle), the injected mice were subjected to CPA procedure as the experimental timeline (Fig. 5d). The mice were randomly divided into two groups: the SS group and the MN group. The results showed that the mice in the MN group exhibited a strong aversion to the withdrawal-paired compartment, whereas mice in the SS group did not exhibit a significant aversion to either compartment (two-way ANOVA, drug treatment factor, $F_{(1,14)}$ = 69.12, $p < 0.0001$; test condition factor, $F_{(1,14)}$ = 40.47, $p < 0.0001$; drug treatment × test condition, $F_{(1,14)}$ = 45.20, $p < 0.0001$, Fig. 5f). After the behavioral assay, mice in each group were sacrificed and slices containing the LH were prepared for whole-cell patch-clamp recording in 30 min after the post-test. Paired-pulse ratio (PPR ratio) of light-evoked inhibitory postsynaptic current (IPSCs) was used as the index of presynaptic GABA release[45]. The result showed that PPR in the MN group was 0.56 ± 0.04, which was significantly lower than that in the SS group (0.99 ± 0.03) (Unpaired $t$ test, $p < 0.0001$. Figure 5g, right). This result suggests that context may induce more GABA release from NAcC D1-MSNs terminals at presynaptic site of LH GABAergic neurons in morphine withdrawn mice.

We further studied whether LH local GABAergic neurons had an inhibitory control on glutamatergic neurons in the LH. AAV-DIO-mCherry was injected into the LH of $Vglut2$-cre mice to label glutamatergic neurons and AAV-mDlx-ChR2-EGFP was injected into the same site to activate GABAergic neurons by following photostimulation (Fig. 6a, left). Four weeks after the injection of virus, when LH GABAergic neurons expressed enough ChR2-EGFP and glutamatergic neurons expressed mCherry (Fig. 6a, right), whole-cell patch-clamp recording was performed in LH glutamatergic neurons (Fig. 6a, right). The result showed that single pulses of photostimulation (blue light, 470 nm, 2 ms) reliably elicited IPSCs in most LH glutamatergic neurons and this IPSCs could be blocked by PTX (Fig. 6b). This result suggests that LH local GABAergic neurons have an inhibitory control on glutamatergic neurons in the LH.

To study whether NAcC D1-MSNs-induced increase in action potential firing of LH glutamatergic neurons was via LH local GABAergic neurons, we examined the influence of "closing" LH local GABAergic neurons using optogenetic methods on NAcC D1-MSNs-induced increase in action potential firing of LH glutamatergic neurons in mice. We used the $D1$-Cre::$Vglut2$-Flpo mice combined with virus injection to specifically label different neurons. AAV-DIO-ChR2-mCherry was injected into the NAcC to activate the D1-MSNs by following photostimulation and AAV-fDIO-mCherry was injected to the LH to label the LH glutamatergic neurons. In order to inhibit LH GABAergic neurons by optogenetic method, a virus encoding the

Natronomonas pharaonis halorhodopsin (NpHR) (AAV-mDlx-eNpHR3.0-EGFP) was injected into the LH (Fig. 6c, left). Four weeks after the injection of virus, whole-cell patch-clamp recording was performed in LH glutamatergic neurons (Fig. 6d, left) when D1-MSNs terminals from the NAcC expressed enough ChR2-mCherry, LH glutamatergic neurons expressed mCherry, and LH local GABAergic neurons expressed mDlx-eNpHR3.0-EGFP (Fig. 6c, right). Top-middle panels in Fig. 6d showed typical action potential trace in response to 100 pA depolarizing current under light-off condition and blue light-on condition. From these raw traces, we could see that in response to this depolarizing current, neurons under blue light-on condition fired more action potentials than that under light-off condition. The average frequency of action potential firing was 8.47 ± 0.54 Hz under light-off condition and 10.88 ± 0.53 Hz under blue light-on condition, showing a significant increase after blue light-on (Paired $t$ test, $p < 0.0001$. Figure 6e, left panel). To study the role of LH local GABAergic neurons in blue light-on-induced increase in the frequency of action potential firing of LH glutamatergic neurons, we examined the influence of the inhibition of LH local GABAergic neurons using yellow light (593.5 nm). Bottom of Fig. 6d shows typical action potential traces in response to 100 pA depolarizing current under blue light-off, yellow light-on, yellow and blue light-on conditions. From these raw traces, we could see that in response to the depolarizing current, LH glutamatergic neurons under yellow light-on condition fired more action potentials than that under blue light-off condition. The average frequency of action potential firing of LH glutamatergic neurons was 6.76 ± 0.61 Hz under blue light-off condition and 9.53 ± 0.64 Hz under yellow light-on condition, showing a significant increase after yellow light-on (one-way ANOVA, $F_{(2, 51)}$ = 8.385, $p = 0.0007$. Tukey's multiple comparisons test, blue light-off group vs. yellow light-on group, $p = 0.0076$. Figure 6e, right panel). But under yellow light-on condition, blue light-on could not further increase the frequency of action potential firing of LH glutamatergic neurons. Under yellow and blue light-on, the average frequency of action potential firing of LH glutamatergic neurons was 10.14 ± 0.62 Hz, showing no further significant change compared to that with only yellow light-on (9.53 ± 0.64 Hz) (one-way ANOVA, $F_{(2, 51)}$ = 8.385, $p = 0.0007$. Tukey's multiple comparisons test, yellow light-on group vs. yellow and blue light-on group, $p = 0.7676$. Figure 6e, right panel). These results indicate that after removing the inhibitory control of LH local GABAergic neurons on LH glutamatergic neurons, the activation of NAcC D1-MSNs projecting to the LH does not increase action potential firing of LH glutamatergic neurons, suggesting that NAcC D1-MSNs projecting to the LH activate LH glutamatergic neurons by inhibiting LH local GABAergic neurons during context-induced expression of morphine withdrawal memory.

## Discussion

The main findings of the present study are that during context-induced expression of morphine withdrawal memory, LH glutamatergic neurons play an important role; NAcC D1-MSNs projecting to the LH is an

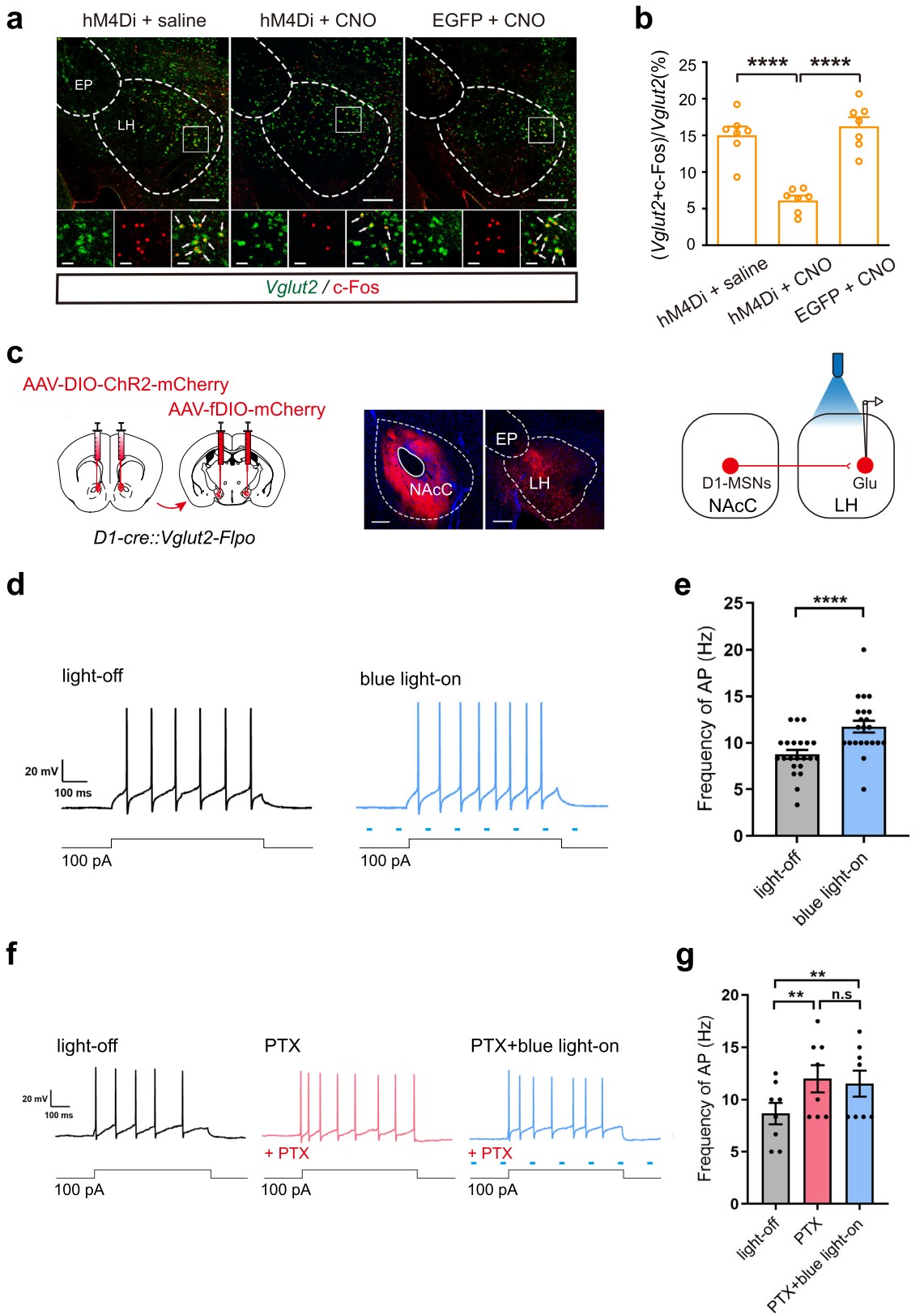

important upstream circuit of the activation of LH glutamatergic neurons; NAcC D1-MSNs projecting to the LH activate LH glutamatergic neurons by removing the inhibitory effect of local GABAergic neurons on LH glutamatergic neurons.

Single-cell transcriptomic analysis shows that the LH contains a large population of glutamatergic neuron[2]. LH glutamatergic neurons

are implicated in physiological events in a variety of innate behaviors, such as feeding, defensive and aversive behaviors. Normal feeding balance relies on appropriate synergism of LH glutamatergic and GABAergic neurons[46]. In defensive behaviors, trimethylthiazole (TMT), the odor of mice predator, significantly increased the activity of LH glutamatergic neurons[47]. Selective activation of LH glutamatergic

**Fig. 4 | The infulence of manipulation of D1-MSNs projecting from NAcC to LH on the activity of LH glutamatergic neurons. a** Top: the *Vglut2* and c-Fos co-labeling neurons in LH in hM4Di + saline, hM4Di + CNO and the EGFP + CNO groups. Magnified image shows the boxed area. Scale bar: 200 μm. Bottom: left, *Vglut2* neurons; middle, c-Fos positive neurons; bottom, c-Fos and *Vglut2* co-labeling neurons. Scale bar: 30 μm. **b** The average percentage of co-labeling of c-Fos and *Vglut2* neurons relative to *Vglut2* neurons in the LH in hM4Di + saline, hM4Di + CNO and the EGFP + CNO groups ($n = 7$ mice in each group). One-way ANOVA, $F_{(2, 18)} = 30.07$, $p < 0.0001$. Tukey's multiple comparisons: hM4Di + saline group vs. hM4Di + CNO group: $p < 0.0001$; hM4Di + saline group vs. EGFP + CNO group: $p = 0.6733$; hM4Di + CNO group vs. EGFP + CNO group: $p < 0.0001$. **c** Left: the diagram of virus injection into the bilateral NAcC (AAV-DIO-ChR2-mCherry) and LH (AAV-fDIO-mCherry) in *D1-Cre::Vglut2-Flpo* mice. Middle: the expression of DIO-ChR2-mCherry in NAcC, the expression of fDIO-mCherry in the LH (Scale bar, 200 μm). Right: the schematics of the whole-cell patch recording approach. **d** The representative AP in LH glutamatergic neurons by the current or by the light stimulation (blue light, 470 nm, 2 ms). **e** The average frequency of AP in LH glutamatergic neurons in the light-off and blue light-on group ($n = 22$ cells). Paired *t* test, $p < 0.0001$. **f** The representative AP in LH glutamatergic neurons by the current or before and after the light stimulation (blue light, 470 nm, 2 ms) in the presence of PTX. **g** The average frequency of AP in LH glutamatergic neurons in the light-off, PTX and PTX + blue light-on groups ($n = 8$ cells). One-way ANOVA, $F_{(7, 14)} = 20.47$, $p < 0.0001$. Tukey's multiple comparisons: light-off group vs. PTX group: $p = 0.0094$, light-off group vs. PTX + blue light-on group: $p = 0.0097$, PTX group vs. PTX + blue light-on group: $p = 0.1953$. Mean ± SEMs. **$p < 0.01$, ***$p < 0.001$, ****$p < 0.0001$.

neurons with optogenetics promoted a series of defense-related behaviors[47] and an aversion in the real-time place avoidance (RTPA) experiments[48]. Moreover, optogenetic stimulation of LH-LHb glutamatergic fibers also produced RTPA[46]. One finding of this study is that under morphine dependence, LH glutamatergic neurons participate in context-induced expression of morphine withdrawal memory.

There is evidence that a part of LH glutamatergic neurons co-release orexin and glutamate from their axonal terminals[1,2,21]. An interesting question is to what extent the involvement of LH glutamatergic neurons in context-induced expression of morphine withdrawal memory is attributed to either glutamate or orexin neurotransmission. The site we have studied in this work is LH rostral site where, based on reports, has a dense distribution of glutamatergic neurons with a sparse distribution of orexin neurons[49–52]. This evidence appears to support that glutamate released from these LH neurons may make a major contribution to context-induced expression of morphine withdrawal memory, whereas orexin released from these LH neurons may do less.

The LH is innervated by multiple structures, such as the medial prefrontal cortex, the lateral habenular nucleus, the basolateral amygdala (BLA), the ventral tegmental area, the lateral septum (LS), the ventral bed nucleus of the stria terminalis (vBNST) and the NAc[1,17,53]. Afferents from the BLA have been demonstrated to involve defensive behaviors[47] and afferents from the LS and the vBNST have been shown to participate in context-induced expression of reward memory[17]. Another finding of this study is that afferents from the NAc constitute an upstream circuit that activates LH glutamatergic neurons to participate in context-induced expression of morphine withdrawal memory.

Originally, ever since it was proposed, 40 years ago, that the NAc played an important role in reward[54,55] and the function of dopamine in the NAc had been central themes in drug abuse studies[54]. Later, accumulating evidence suggested that the NAc also participated in drug withdrawal[37]. In addition, the NAc is also involved in the formation and expression of drug addiction memory, including reward memory and withdrawal memory[36,56]. The role of the NAc in different stages of drug addiction may be related to differential activity of NAc neurons in its two subregions, NAcSh and NAcC, where there are two populations of medium spiny projection neurons (MSNs) (MSNs): D1-MSNs and D2-MSNs[33,34]. Previous studies showed that the activity of different MSNs in NAcSh was closed to different stages of drug addiction. For example, for rewarding effect, cocaine could induce an increase in the expression of c-Fos in D1-MSNs of NAcSh[57]. Using optogenetic method, the activation of D1-MSNs of NAcSh enhanced the rewarding effects of cocaine, but the activation of D2-MSNs of NAcSh antagonized cocaine reward[35]. For the expression of morphine withdrawal memory, the activation of D2-MSNs of NAcSh contributed to context-induced expression of morphine withdrawal memory[36]. However, the NAcSh is only one subregion of the NAc and the role of different MSNs of another subregion of the NAc, the core region, in different stages of drug addiction remains unknown. The present

results suggest that NAcC D1-MSNs, rather than NAcC D2-MSNs, contribute to context-induced expression of morphine withdrawal memory. This result combined with previous study[36] indicates that when context activates the NAc to induce the expression of morphine withdrawal memory, it may be through two neural circuits: one is shell D2-MSNs-mediated neural circuit and another is core D1-MSNs-mediated neural circuit.

It has been known that D1-MSNs in the NAcC are GABAergic neurons[43]. So, it is not possible that D1-MSNs in the NAcC activate LH glutamatergic neurons by a direct innervation approach. Moreover, using the whole-cell patch-clamp method combined with the optogenetic method, our result showed that the direct innervation from the NAcC D1-MSNs to LH glutamatergic neurons was very rare. Another possible approach for D1-MSNs in the NAcC to activate LH glutamatergic neurons is that D1-MSNs may produce an inhibitory effect on local GABAergic neurons and then induce a disinhibitory effect on glutamatergic neurons in the LH. However, whether there is such a disinhibitory circuit from NAcC D1-MSNs to LH glutamatergic neurons remains unknown. Our results showed that (1) single pulses of photostimulation (blue light) of NAcC D1-MSNs could elicit IPSCs in most LH GABAergic neurons and this IPSCs could be blocked by PTX, suggesting that NAcC D1-MSNs innervated LH GABAergic neurons; (2) single pulses of photostimulation (blue light) of LH local GABAergic neurons reliably elicited IPSCs in most LH glutamatergic neurons and this IPSCs could be blocked by PTX, suggesting that LH local GABAergic neurons innervated glutamatergic neurons in the LH; (3) single pulses of photoinhibition (yellow light) of LH local GABAergic neurons increased firing of action potentials, suggesting there was an inhibitory control of LH local GABAergic neurons on glutamatergic neurons of the LH; (4) after removing an inhibitory control of LH local GABAergic neurons on glutamatergic neurons of the LH using photoinhibition (yellow light), the increasing effect of photostimulation (blue light) of NAcC D1-MSNs on firing of LH glutamatergic neurons disappeared, suggesting that NAcC D1-MSNs projecting to the LH activated LH glutamatergic neurons by inhibiting LH local GABAergic neurons. These results reveal the presence of a disinhibition circuit between NAcC D1-MSNs and LH glutamatergic neurons and this disinhibitory circuit mediates the activating effect of D1-MSNs on LH glutamatergic neurons during context-induced expression of morphine withdrawal memory. In addition, our result showed that context could induce more GABA release from NAcC D1-MSNs terminals at presynaptic site of LH GABAergic neurons in morphine withdrawn mice, compared to normal mice. The mechanisms underlying this increase in GABA release from NAcC D1-MSNs projecting to the LH following CPA memory expression remains unknown. We speculate that it may be related to contex-withdrawal association-induced strengthening of D1-MSNs projecting to the LH because in our previous study, we observed contex-withdrawal association-induced strengthening in other projection neurons[58].

In conclusion, the present study suggests that disinhibited LH glutamatergic neurons by neural circuits from the NAcC

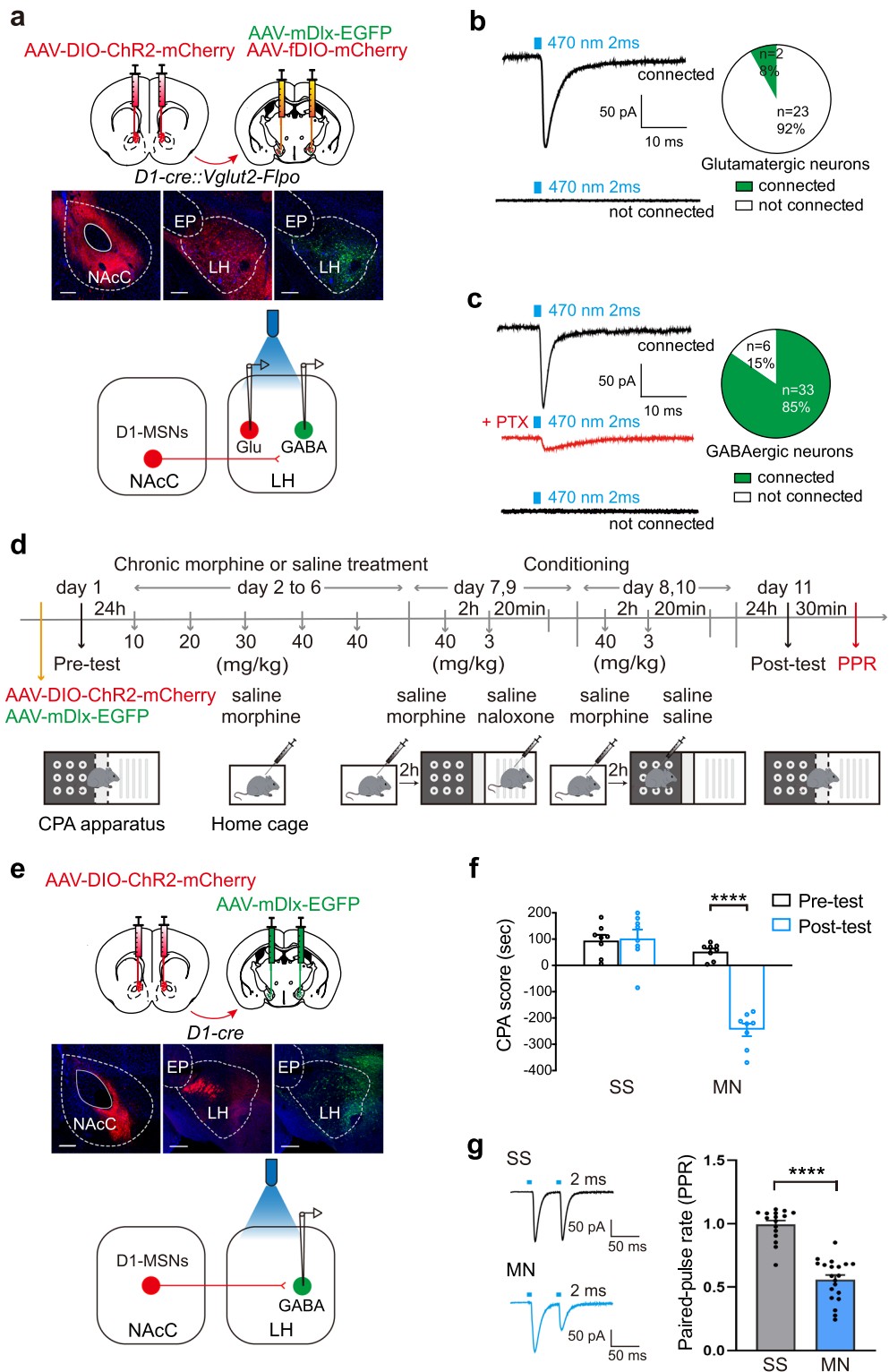

importantly contribute to context-induced expression of morphine withdrawal memory.

## Methods

### Animals

In our experiments, we used male adult (8–12 weeks) C57BL/6J wild-type mice and four kinds of transgenic mice: *D1-cre* mice, *D2-cre* mice, *Vglut2-cre* mice, *D1-Cre::Vglut2-Flpo* mice. The *D1-cre* (#034258-UCD) and *D2-cre* mice (#032108-UCD) were from the Mutant Mouse Resource and Research Center. The *Vglut2-Cre* mice (Vglut2, vesicular glutamate transporter 2; #016963) was from Jackson Laboratory. The *D1-Cre::Vglut2-Flpo* mice was gifted from Dr. Wei L. Shen. All transgenic mice were bred onto C57BL/6 background for at least one generation. All experimental procedures involving mice were housed in a 12 h light/dark cycle in a temperature-and humidity-controlled environment with food and water freely available. All experimental procedures conformed to Fudan University as well as international guidelines on the ethical use of animals. All efforts were made to minimize animal suffering and reduce the number of animals used.

**Fig. 5 | The projection of NAcC D1-MSNs to LH and the infulence of morphine withdrawal memory on this projection. a** Top: the diagram of virus injection into the NAcC (AAV-DIO-ChR2-mCherry) and LH (AAV-mDlx-EGFP and AAV-fDIO-mCherry) in *D1-Cre::Vglut2-Flpo* mice. Middle: the expression of ChR2-mCherry in NAcC, the expression of mDlx-EGFP and fDIO-mCherry in LH (Scale bar, 200 μm). Bottom: the schematics of the whole-cell patch recording. **b** Left: the representative IPSCs evoked in connected and non-connected LH glutamatergic neurons by the light stimulation (blue light, 470 nm, 2 ms). Right: the percentage of connected and non-connected LH glutamatergic neurons ($n$ = 25 cells of 5 mice). **c** Left: the representative IPSCs evoked in connected and non-connected LH GABAergic neurons by light stimulation (blue light, 470 nm, 2 ms). PTX (100 μM) blocked light-evoked IPSCs. Right: the percentage of connected and non-connected LH

GABAergic neurons ($n$ = 39 cells of 5 mice). **d** The experimental timeline. **e** Top: the diagram of virus injection into the NAcC (AAV-DIO-ChR2-mCherry) and LH (AAV-mDlx-EGFP) in *D1-Cre* mice; Middle: the expression of ChR2-mCherry in NAcC and LH, the expression of mDlx-EGFP in the LH (Scale bar, 200 μm). Bottom: the schematics of the whole-cell patch recording approach. **f** The average CPA score in SS ($n$ = 8) and MN ($n$ = 8) groups. Two-way ANOVA, drug treatment factor, $F_{(1,14)}$ = 69.12, $p$ < 0.0001; test condition factor, $F_{(1,14)}$ = 40.47, $p$ < 0.0001; drug treatment × test condition, $F_{(1,14)}$ = 45.20, $p$ < 0.0001. **g** Left: the representative paired-pulse ratio (PPR) of light-evoked (470 nm, 2 ms) IPSCs evoked in LH GABAergic neurons in SS and MN groups. Right: average PPR in SS ($n$ = 16 cells) and MN ($n$ = 20 cells) groups. Unpaired $t$ test, $p$ < 0.0001. Mean ± SEMs. ****$p$ < 0.0001.

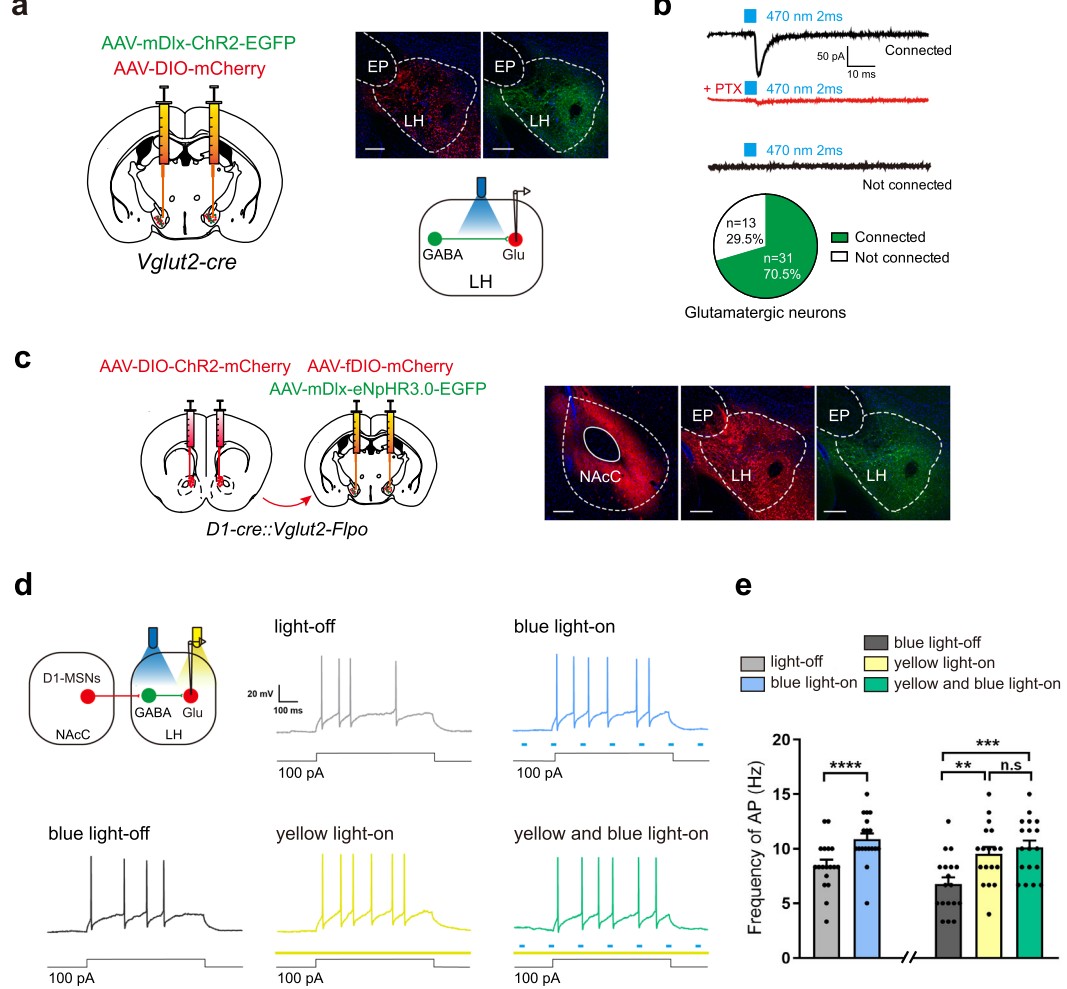

**Fig. 6 | The influence of modulation of LH local GABAergic neurons on the activity of LH glutamatergic neurons induced by the NAcC D1-MSNs projection neurons. a** Left: the diagram of virus injection into the LH (AAV-mDlx-ChR2-EGFP and AAV-DIO-mCherry) in *Vglut2-cre* mice. Upper right: the expression of mDlx-ChR2-EGFP, DIO-mCherry in the LH (Scale bar, 200 μm). Bottom right: the schematics of the whole-cell patch recording. **b** Top: the representative IPSCs evoked in connected and non-connected LH glutamatergic neurons by the light stimulation (blue light, 470 nm, 2 ms). PTX (100 μM) blocked light-evoked IPSCs; Bottom: the percentage of connected and non-connected LH glutamatergic neurons ($n$ = 44 cells of 5 mice). **c** Left: the diagram of virus injection into the NAcC (AAV-DIO-ChR2-mCherry) and LH (AAV-fDIO-mCherry and AAV-mDlx-eNpHR3.0-EGFP) in *D1-*

*Cre::Vglut2-Flpo* mice; Right: the expression of DIO-ChR2-mCherry in NAcC, mDlx-eNpHR3.0-EGFP and fDIO-mCherry in the LH (Scale bar, 200 μm). **d** The schematics of the whole-cell patch recording approach and the representative AP in LH glutamatergic neurons by light stimulation (blue light, 470 nm or yellow light, 593.5 nm) or by light stimulation (yellow light, 593.5 nm + blue light, 470 nm). **e** The average frequency of LH glutamatergic neurons AP in the different groups ($n$ = 18 cells). Left panel, Paired $t$ test, $p$ < 0.0001. Right panel, One-way ANOVA, $F_{(2, 51)}$ = 8.385, $p$ = 0.0007. Tukey's multiple comparisons test, yellow light-on group vs. yellow and blue light-on group, $p$ = 0.7676; blue light-off group vs. yellow light-on group: p = 0.0076. Mean ± SEMs. **$p$ < 0.01, ***$p$ < 0.001, ****$p$ < 0.0001.

## Conditioned place aversion (CPA)

CPA was conducted using a three-compartment place conditioning apparatus (Med Associates, USA) with distinct visual and tactile context, and the procedure was similar to that described previously[59,60].

The CPA procedure included four phases: pre-test (day 1), drug treatment (days 2–6), conditioning (days 7–10), and post-test (day 11).

On pre-test day (day 1), The mice were placed in the middle neutral area and were allowed to freely access both sides of the

apparatus for 15 min. In our study, considering to eliminate more influence of strong unconditioned preference on the results, we discarded mice with strong unconditioned aversion or preference for any compartment (i.e., >80% of the session time) and kept mice with minor preference for one compartment. All eligible mice were randomly divided into four groups: SS, SN, MS and MN.

On drug treatment days (days 2–6), mice were treated by twice daily intraperitoneal injections of morphine (Northeast pharmaceutical group Shenyang, China) in MS and MN groups. Morphine doses were progressively increased over 5 days from 10 mg/kg to 40 mg/kg: day 1, 10 mg/kg; day 2, 20 mg/kg; day 3, 30 mg/kg; days 4 and 5, 40 mg/kg. Mice in SS and SN groups were treated with saline. All morphine or saline injections were given in the animal's home cage in the pretreatment mode.

On conditioning days (days 7 and 9), mice in the MN group were injected with Naloxone (3 mg/kg, intraperitoneally; #S3066, Selleckchem) 2 hours after receiving morphine injection (40 mg/kg, intraperitoneally) to induce withdrawal and confined in its morphine withdrawal-paired compartment (minor preference compartment during pre-test) for 20 min. On conditioning days (days 8 and 10), the mice in the MN group was injected with saline injection (0.1 ml, intraperitoneally) 2 h after receiving morphine injection (40 mg/kg, intraperitoneally) and confined in its saline-paired compartment (opposite compartment) for 20 min.

On post-test day (day 11), the mice were allowed to freely exploring three compartments for 15 min. CPA score was defined as the time in the minor preferred compartment minus the time in the opposite compartment. Since the withdrawal conditioned training was performed in the minor preference compartment, the opposite compartment became more preferred compartment after twice withdrawal training. Therefore, the CPA score was positive in the pre-test session and negative in the post-test of conditioned withdrawal group. All the CPA procedures with three control groups (SS, SN and MS groups) were the same as the MN group, except the mode of drug administration (Fig. 1a).

For in vivo chemogenetic (Designer Receptors Exclusively Activated by Designer Drugs, DREADD) inhibition in CPA experiments, CNO (1 mg/kg, #S6887, Selleckchem) or saline was injected 40 min before the post-test.

### Stereotactic surgery

Mice were anesthetized with pentobarbital (100 mg/kg sodium pentobarbital, intraperitoneally (i.p.))[61–63] before the stereotaxic surgery was performed. Stereotactic surgeries to inject the virus into the LH and NAcC were performed on a RWD stereotactic frame (model 68507, RWD Life Science). Briefly, the skull was exposed with a small incision and holes were drilled to deliver virus with glass electrode. According to the mouse brain atlas[64], we determined the coordinates of viral injection. The coordinates of bregma [anteroposterior (AP): +1.60 mm; dorsoventral (DV): −4.60 mm; mediolateral (ML): ±1.21 mm] were used to target NAcC bilaterally, and the coordinates of bregma (AP: −1.00 mm, DV: −5.12 mm, ML: ±1.20 mm) were used to target LH bilaterally. Injection of 300 nl virus per side were made with a hydraulic pump at a speed of 40 nl per minute. For the stereotaxic injections, glass electrode was retained in the target region for an additional 10 min to allow diffusion of the injected virus. Viral-injected mice were allowed at least 3 weeks after the surgery to recover and to express the virus before CPA behavioral tests or electrophysiological recordings. To ensure the accuracy of virus injection sites, animals were sacrificed and slices containing the regions of interest were prepared. The brain slices were imaged with a 10 × objective using a microscope to observe the site of virus fluorescent expression. If the expression of virus is not within the region of interest, the data of this animals was excluded from further analysis. In our study, 11 out of 178 animals were excluded for further

analysis because of the mistargeting of virus.Details are in the Table 1 of supplementary materials.

For in vivo DREADD inhibition of LH glutamatergic neurons in CPA experiments, *Vglut2-cre* mice were injected with AAV2/9-EF1α-DIO-hM4D(Gi)-EGFP or AAV2/9-EF1α-DIO-EGFP ($5.83 \times 10^{12}$ vector genomes/ml, BrainVTA, China) bilaterally into the LH. For in vivo DREADD inhibition of D1-MSNs projections from NAcC to LH in CPA experiments, *D1-cre* mice were injected with AAV2/9-nEF1α-fDIO-hM4D(Gi)-EGFP or AAV2/9-nEF1α-fDIO-EGFP ($2.15 \times 10^{12}$ vector genomes/ml, BrainVTA, China) bilaterally into the NAcC, and AAV2-Retro-CAG-FLEX-Flpo ($3.18 \times 10^{12}$ vector genomes/ml, Taitool Bioscience, China) into the LH. For in vivo DREADD inhibition of NAcC D1-MSNs and D2-MSNs, *D1-cre* mice and *D2-cre* mice were injected with AAV2/9-hEF1α-DIO-hM4D(Gi)-mCherry or AAV2/9-hEF1α-DIO-mCherry to NAcC ($3.36 \times 10^{12}$ vector genomes/ml, Taitool Bioscience, China) bilaterally into the NAcC. For retrograde labeling experiments, wild-type mice received injections of CTB647 (1 μg/μl, #C34778, Invitrogen) bilaterally into the LH. For optogenetic activation of D1-MSNs projections from NAcC to LH glutamatergic neurons experiments, *D1-Cre::Vglut2-Flpo* mice were injected with AAV2/9-hEF1α-DIO-hChR2-mCherry ($4.58 \times 10^{12}$ vector genomes/ml, Taitool Bioscience, China) into NAcC. For labeling LH glutamatergic neurons, we injected the AAV2/9-hEF1α-fDIO-mCherry ($4.12 \times 10^{12}$ vector genomes/ml, Taitool Bioscience, China) into LH in *D1-Cre::Vglut2-Flpo* mice, and injected the AAV2/9-hEF1α-DIO-mCherry ($1.36 \times 10^{12}$ vector genomes/ml, Taitool Bioscience, China) into LH in *Vglut2-cre* mice. For labeling LH GABAergic neurons. we injected the AAV2/9-mDlx-EGFP ($1.45 \times 10^{12}$ vector genomes/ml, Taitool Bioscience, China) into LH. For optogenetic modulation of LH GABAergic neurons, we injected the AAV2/9-mDlx-eNpHR3.0-EGFP ($1.68 \times 10^{12}$ vector genomes/ml, Taitool Bioscience, China) or AAV2/9-mDlx-hChR2-EGFP ($2.19 \times 10^{12}$ vector genomes/ml, Taitool Bioscience, China) into LH.

### In vitro optogenetic approach for electrophysiology

Mice were anesthetized with pentobarbital (100 mg/kg, i.p.) and then were euthanized by exsanguination. The brain was removed rapidly from the skull and placed in modified artificial cerebrospinal fluid (ACSF) containing 75 mM sucrose, 88 mM NaCl, 2.5 mM KCl, 1.25 mM NaH$_2$PO$_4$, 7 mM MgCl$_2$, 0.5 mM CaCl$_2$, 25 mM NaHCO$_3$, and saturated with 95% O$_2$ and 5% CO$_2$ at ~0 °C. Coronal 250 μm slices containing LH were cut on a vibratome (VT-1200, Leica, Wetzlar, Germany) and transferred to normal ACSF containing 126 mM NaCl, 2.5 mM KCl, 1.25 mM NaH$_2$PO$_4$, 2 mM MgSO$_4$, 2.5 mM CaCl$_2$, 25 mM, NaHCO$_3$, and 10 mM glucose at 32 °C. Slices were incubated for at least 60 min before patch-clamp recording.

LH neurons were visualized on an upright microscope (BX50WI, Olympus, Japan) using infrared differential interference contrast or fluorescent optics. Whole-cell current and voltage-clamp recordings were recorded using an EPC10 amplifier and Patch Master 2.54 software (HEKA, Lambrecht, Germany). Electrodes had a resistance of 3–4 MΩ when filled with the patch pipette solution. The internal pipette solution contained for action potential (AP): 135 mM K-gluconate, 4 mM KCl, 10 mM HEPES, 10 mM sodium phosphocreatine, 4 mM Mg-ATP, 0.3 mM Na$_3$-GTP (pH 7.2, 276 mOsm). The internal pipette solution contained for IPSCs and PPR: 130 mM KCl, 8 mM NaCl, 0.1 mM CaCl$_2$, 0.6 mM EGTA, 2 mM Mg-ATP, 0.1 mM Na$_3$-GTP and 10 mM HEPES (pH 7.2, 276 mOsm). Cells were held at −70 mV under a voltage-clamp mode to record. The channelrhodopsin (ChR2) was stimulated by the 470 nm blue light (2 ms pulses) and the eNpHR3.0 was stimulated by the 593.5 nm yellow light, which delivered via an optical fiber (core diameter 200 μm, NA = 0.39, ThorLabs) coupled to a LED light source (Mightex) 500 μm above the recording cell. To verify whether the changes of IPSCs and AP caused by GABA neurotransmitters, we used picrotoxin (PTX, 100 μM, Sigma-Aldrich) to block the GABA-A receptors. Cells were held at 0 pA under a current-clamp mode to

record current injected action potential (100 pA, 600 ms). For recording spontaneous firing, neurons were continuously holding at 10–20 pA to induce evident spontaneous firing.

## Immunohistochemistry and in situ hybridization

For immunohistochemistry experiments, 90 min after the post-test, mice were anesthetized with pentobarbital (100 mg/kg, i.p.) and perfused with 0.9% saline, followed by ice-cold solution of 4% paraformaldehyde (PFA) in phosphate-buffered saline (PBS, pH 7.4). The brains were removed and fixed in 4% PFA overnight. Next, the brains were cut in 40 μm coronal sections using a vibratome (Leica, USA) and collected in PBS. Brain sections were first washed in PBS (3 × 10 min), then blocked at 4 °C with 10% normal goat serum, 0.3% Triton X-100 (PBS) and then incubated with guinea pig anti−c-Fos antibody (#226004, Synaptic Systems, Germany) diluted 1:1000 overnight at 4 °C. Then, sections were rinsed in PBS for three times (5 min for each wash) and incubated with goat anti-guinea pig immunoglobulin G (IgG) antibody (Vector, USA) diluted 1:500 for 1 h at 25 °C, followed by Streptavidin-Cy3™ (#S6402, Sigma-Aldrich, USA) diluted 1:1000 for 1 h at 25 °C. Subsequently, they were rinsed in PBS for three times (5 min for each wash).

For in situ hybridization, DNA templates for generating in situ probes were cloned using the following primer sets for each corresponding gene: Vglut2, 5′-ATCGACTAGTCCAAATCTTACGGTGCTA CCTC-3′ and 5′-ATCGCTCGAGTAGCCATCTTTCCTGTTCCACT-3′; Drd1(D1), 5′-CTCATAAGCTTTTACATCCCCG-3′ and 5′- GAGACATCGGT GTCATAGTCCA-3′. Anti-sense RNA probes were transcribed with T7 RNA polymerase (#P207E, Promega, USA) and digoxigenin (DIG)-labeled nucleotides. Mice were anesthetized and transcardially perfused with 0.1% DEPC-treated PBS (D-PBS) followed by ice-cold 4% PFA in D-PBS. After that, brains were sectioned at 40 μm thickness using a vibratome (Leica, USA). Brain sections were washed in D-PBS containing 0.5% $H_2O_2$ for 30 min, 2× SSC buffer containing 0.1% triton for 30 min, acetylated in 0.1 M triethanolamine (pH 8.0) with 0.25% aceticanhydride for 10 min, equilibrated in prehybridization solution for 2 h at 65 °C and subsequently incubated with 0.5 μg /ml of specific RNA probes in hybridization buffer overnight at 65 °C. The next day, sections were rinsed in prehybridization solution and pre-hyb/TBST (TBS with 0.1% tween-20) for 30 min each. Next, sections were washed with TBST for twice and 1XTAE for three times, each for 5 min. Sections were then transferred into wells in 2% agarose gel, which were run in 1XTAE at 60 V for 2 h to remove unhybridized probes. Sections were then washed twice in TBST, and subsequently incubated with sheep anti-digoxygenin-POD (1:500, Roche) and guinea pig anti−c-Fos antibody (1:500 dilution) for co-staining purposes, in 0.5% blocking reagent (Roche) at 4 °C overnight. On the third day, sections were washed with TBST for three times, then staining with TSA Fluorescein detection kit (TSA Fluorescein, PerkinElmer). For fluorescent in situ hybridization combined with immunohistochemistry, then sections were incubated with goat anti-guinea pig immunoglobulin G (IgG) antibody, followed by Streptavidin-Cy3™, finally were stained with DAPI (5 mg/ml, 1:1000, #D9542, Sigma-Aldrich) in PBS. All sections were washed after staining and mounted on glass slides.

Images were obtained by confocal microscopy with a 20× objection len and collected at a resolution of 1024 × 1024 pixels. Quantification of c-Fos, labeled neurons was performed with Image J software with the same threshold. The positive cells were defined with staining above basal background. Counts collected from at least 4 sections from each mouse were averaged to produce value.

## Single-cell RT-PCR

After whole-cell patch-clamp recordings slices containing LH, non-labeled neurons were harvested into patch pipettes. Specific target reverse transcription and amplification was performed using Single-Cell Sequence Specific Amplification Kit (Vazyme, China) following the manufacturer's instructions. We conducted the reverse transcription and pre-amplification in 5 μl of RT-Pre Amp Master Mix (Vazyme, China). The protocol included reverse transcription at 50 °C for 60 min and initial 3 min denaturizing step at 95 °C followed by 17 cycles of 15 s denaturation at 95 °C, 15 min annealing and elongation at 60 °C. The second round of amplification was performed on the mastercycler ep realplex (Eppendorf) using the Hieff qPCR SYBR Green Master Mix (Yeasen, China) according to the manufacturer's instructions (95 °C for 5 min followed by 40 cycles at 95 °C for 10 s, 56 °C for 20 s, and 72 °C for 20 s). The final products were run on a Gelgreen staining 3% agarose gel for visualization under ultraviolet light. Primers for Vglut2 were designed according to NCBI published sequences.

List of primer sequences used for RT-PCR.

| Target gene | Size(bp) | | Primers | Genbank No. |
|---|---|---|---|---|
| Vglut2 | 132 bp | Forward | AGACCCTGAGGAAACAAGCG | NM_080853.3 |
| | | Reverse | TCCTGTGAGGTAGCACCGTA | |

## Statistical analysis

Statistical significance was determined using Student's t test for comparisons between two groups or analyzes of variance (ANOVAs) for comparisons among three or more groups. The normality test of the data was performed by Shapiro−Wilk test and the homoscedasticity was performed by F test. The nonnormalized data were analyzed with nonparametric test. One-way ANOVA was followed by Tukey's multiple comparisons test and two-way repeated ANOVA was followed by Bonferroni's multiple comparisons test to calculate p values (treatment with different drugs as the between-subject factors and test condition as the within-subjects factor). In all cases, n refers to the number of cells or animals. Offline data analysis was performed using a PatchMaster (HEKA). Graphpad Prism 8.4 was used to process and analyze data and make statistical graphs. Data are presented as Means ± SEMs.

Experiments shown in Figs. 1f; 2b, d; 3b, d, h; 4c; 5a, e; 6a, c and Supplementary Fig. 1i. were repeated independently in at least four animals with similar results. We choose a representative diagram to display the injection sites and viral expressions.

## Reporting summary

Further information on research design is available in the Nature Portfolio Reporting Summary linked to this article.

## Data availability

The authors declare that all data supporting the findings of this study are available within the article and its supplementary information file, and are available from the corresponding author upon request without restrictions. Source data are provided with this paper.

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

## Acknowledgements

This work was supported by the Major Project of the Science & Technology Innovation 2030 of China (STI2030-Major Projects 2021ZD0203500 to P.Z.), the project of Foundation of National Natural Science of China (32030051 to P.Z., 31970956 to B.L., 32171025 to M.C.) and the Shanghai Municipal Science and Technology Major Project (No. 2018SHZDZX01 to P.Z.), ZJ Lab and Shanghai Center for Brain Science and Brain-Inspired Technology to P.Z.

## Author contributions

P.Z. conceived the study. M.C., B.L., and H.S. designed the experiments. H.S., C.L., and Y.Y. analyzed the data. H.S., and C.L. conducted stereotactic surgery, behavior and immunohistochemistry experiments. H.S., and Y.Y. conducted patch-clamp recording experiments. P.Z., H.S., and C.L. wrote the manuscript. Y.F., D.C., L.Y., D.S., and Z.X.C. helped maintaining the mice colony. H.Y., and X.G. prepared the internal pipette solution for electrophysiology. C.C., Y.W., and Z.Y.C. prepared all the reagents for immunohistochemistry and in situ hybridization. All authors reviewed the manuscript.

## Competing interests

The authors declare no competing interests.
