## [Peer Review File · Nature Communications]

Nucleus accumbens circuit disinhibits lateral hypothalamus glutamatergic neurons contributing to morphine withdrawal memory in male miceREVIEWER COMMENTS

Reviewer #1 (Remarks to the Author):

I have reviewed with eager interest the manuscript entitled “Disinhibited lateral hypothalamus glutamatergic neurons by circuits from nucleus accumbens contribute to withdrawal memory retrieval” by Huan Sheng et al. The manuscript shows that a specific population of LH glutamate neurons is FOC-activated during the retrieval of morphine withdrawal memory in mice previously conditioned to a context where they experienced negative affectivity of naloxone-precipitated withdrawal. Chemogenetic inhibition of glutamate neurons blocked the CPA. The neurons are indirectly activated by D1R-expressing GABAergic MSNs that project from the core region of the nucleus accumbens (NACC) to the LH. The MSNs are electrophysiologically coupled to LH GABA neurons, through which they activate LH glutamate neurons in a disinhibition mode. The whole functionality of the path was further confirmed by showing that blocking LH GABA neurons in the time of optogenetic stimulation of NACC MSNs abolished LH glutamate neuron responsivity. In my view, the methodological aspect of the work is remarkably rich and the results are outstanding. There are some comments, however, that may help improve the manuscript.

Conceptual issues:

1- As mentioned in the manuscript, there is evidence that LH glutamate neurons colocalize with orexin. The two neurotransmitters are not necessarily released always together as it has been shown that low-frequency photostimulation of orexin neurons caused release of glutamate, whereas high-frequency stimulation caused release of glutamate and orexin (Patricia Bonnavion, 2016). The question now is to what extent the involvement of LH glutamate neurons in naloxone-induced CPA is attributed to either glutamate or orexin neurotransmission? Although chemogenetic inhibition of glutamate neurons blocked CPA, this cannot dissociate glutamate effect from possible orexin effect. The Introduction provided some evidence that orexin may enhance glutamate neurotransmission, however, orexin has been implicated in the rewarding properties of drug.

2- Fig 1d. c-Fos immunoreactivity was observed only in 10% of LH glutamate neurons in naloxone-induced CPA group beyond those happened in control groups (~18% vs. ~8%). The same result came out of experiments related to Fig 3b. Although significant statistically, is it also significant biologically? Even if this biological effect was sought to be further confirmed by chemogenetic inhibition of glutamate neurons, this procedure probably dropped out virtually all glutamate neurons (see Fig 1f) not only the specific subset that was c-Fos activated during CPA.

3- LH contains both GABA interneurons and GABA projection neurons. The latter that projects to the VTA is rewarding when stimulated (Jennings, 2015). Therefore, its inhibition may have anti-reward effect. How can the authors exclude the possibility that D1-MSNs inhibit this population of LH GABA neurons (rather than GABA interneurons) and this way they cause CPA during morphine withdrawal?

Introduction:

4- Previous indications regarding the involvement of LH glutamate neurons in aversive behavior may help justify the significance of studying their role also in withdrawal memory. For example, consider the

following references: Jennings, 2013; Stamatakis 2016.

Results:

- 5- Fig 1g/line 167. Within-group comparisons are needed. Was CPA blocked COMPLETELY by chemogenetic inhibition of LH glutamate neurons?
- 6- Fig 2d. c-Fos expression is not evident; especially in the Mor+Nal group it seems that expression is not that different from other control groups. Please consider replacing the representative figures or repeating the experiments or confirming the results by western blotting.
- 7- Fig 3d. More direct data needs to be provided showing that photostimulation of NACC MSNs activated the neurons. Therefore, the response of these neurons in the form of increased firing rates to optogenetic stimulation would be of interest. Alternatively, experiments that show longer duration of photostimulation alone (without simultaneous electrical stimulation) is able to increase firing activity of LH glutamate neurons is required. Please consider providing a raster plot and PSTH for the response of LH glutamate neurons to MSN photostimulation. Providing data from a Light off + PTX condition would be of interest to see the effect of PTX alone. In addition, performing the same experiments in a non-labeled neuron, preferably glutamatergic one, would increase the reliability of the claim.
- 8- Fig 4b, 4C. 8% of glutamate and 85% of GABA neurons were inhibited by D1-MSNs photostimulation. 75% of glutamate neurons were inhibited by photostimulation of GABA neurons. Therefore, ~59% of glutamate neurons should be excited by activation of D1-MSNs. Considering that ~30% of D1-MSNs were Fos-activated by CPA (Fig 2e), it is expected that ~17.7% of LH glutamate neurons should be excited during CPA, which is very close to that found in experiments for Fig 1d (18%) or Fig 3b (15%). Does this estimation make sense?
- 9- Fig 5c. I suppose that the schematic of neuronal connections is not correct.
- 10- Fig 5d. The data from light-off condition is required. However, comparing the baseline firing of LH glutamate neurons in Fig 3d (~4 AP/s) with that of Yellow-light-on condition in Fig 5d (~5 AP/s) shows no significant difference. Why did not LH GABA photoinhibition cause any increase in the firing rates of glutamate neurons? Does it mean LH GABA neurons have no inhibitory effect on LH glutamate neurons in the resting state? If so, how come their inhibition by MSNs give rise to disinhibition of glutamate neurons?

Minor comments:

- 11- Although the manuscript was written in a sufficiently good way, it needs to be revised in terms of English writing.
- 12- Consider rephrasing the title. The word "disinhibited" cannot (or at least seems odd to) come with "by". What about using the singular form of "circuits"?
- 13- Throughout the text, the word "conditioned" can be omitted from the expression "conditioned context-induced retrieval of withdrawal memory." In addition, this expression was often repeated redundantly in the text (for an example see lines 210-214).
- 14- Line 203. Remove "we proposed."
- 15- Line 109. Swap the place of "projecting" and "neurons".
- 16- Line 123. Remove "in morphine withdrawn mice".
- 17- Lines 125-129. Use "were trained to conditioned place aversion" or other expressions instead of

“accepted the conditioned context training”.

18- Throughout the text, use “conditioning training” or other expressions instead of “conditioned context training”.

19- Line 276. To study ...

20- Lines 292-296. The first (“to directly test...”) and last (“to examine...”) parts of the sentence are almost the same, hence redundant.

21- Line 313. “and the activation of LH glutamatergic neurons are” can be removed with no harm to the sentence.

References used in the comments

- Bonnavion P, Mickelsen LE, Fujita A, de Lecea L, Jackson AC. Hubs and spokes of the lateral hypothalamus: cell types, circuits and behaviour. *J Physiol*. 2016 Nov 15;594(22):6443-6462. doi: 10.1113/JP271946. Epub 2016 Jul 19. PMID: 27302606; PMCID: PMC5108896.
- Jennings JH, Rizzi G, Stamatakis AM, Ung RL, Stuber GD (2013) The inhibitory circuit architecture of the lateral hypothalamus orchestrates feeding. *Science* 341:1517–1521.
- Jennings, J.H. et al. Visualizing hypothalamic network dynamics for appetitive and consummatory behaviors. *Cell* 160, 516–527 (2015).
- Stamatakis AM, Van Swieten M, Basiri ML, Blair GA, Kantak P, Stuber GD (2016) Lateral hypothalamic area glutamatergic neurons and their projections to the lateral habenula regulate feeding and reward. *J Neurosci* 36:302–311.

Reviewer #2 (Remarks to the Author):

Reviewer comments

This manuscript have been addressed an interesting hypothesis that the DA1-MSNs of nucleus accumbens core (NAc) to lateral hypothalamus (LH) activate the LH glutamatergic neurons. The present study suggests that the activation of LH glutamatergic neurons plays an important role in the retrieval of morphine withdrawal memory. This activation of LH glutamatergic neurons is initiated by DA1-MSNs of NAc via disinhibiting the local GABAergic neurons of LH. Methods were appropriately conducted. Saline and chronic morphine treatment were given to mice for 2- 6 days with increasing order of dose per day (10, 20, 30, 40, 40, 40 mg/kg, intraperitoneally). Outcome measures behavioral test of conditioned place aversion (CPA) score for the retrieval of withdrawal memory. Statistical results are fully presented in the text for the behavioral experiments. Data presentation is clear and convincing. Results show strong evidence for withdrawal memory retrieval in condition place aversion model, in which CPA score of morphine + naloxone group shows significantly higher than saline + saline, saline + naloxone and morphine + saline group. Appropriate references are cited. The authors are to be complimented on their careful experimental design and well presented data. A few major and minor points should be addressed to further improve this excellent manuscript.

Major Comments:

#Comment 1: In Introduction 6th paragraph, authors have mentioned that D2-MSNs in the shell of the NAc have no direct projections to the LH, references cited by authors does not support to this statement. However, this sentence is seems to be very contradictory to this line i.e. D2-MSNs in the shell of the NAc have direct projections to the LH (Please go through these references, Frazier et al., 2010; Li et al., 2018) and do corrections accordingly.

#Comment 2: In Introduction 2nd paragraph last line, authors are saying that the role of glutamatergic neurons of the LH in drug addiction remains unknown. However, the role of glutamatergic neurons in the LH have been contributed for the drug of abuse (Kindly, refer the article Chen et al., 2013). This is contrary to the statement provided by authors. Please go through the extensive literature survey and update the information.

#Comment 3: In Introduction section, there is no any reference in 4th paragraph. On what basis authors have provided the information, every information should have been included with appropriate citation of references.

#Comment 4: In the method section, authors have not provided the data regarding cannula verification, if authors have done cannula verification; please provide the original histological images and the sample size use in the experiment. It is very much necessary because after performing stereotaxic surgery by using coordinates from atlases reference to bregma, as for many instances cannulae are not placed at right region of interest and leads to mislead of data collection. This also enables to decide the number of animals for the concerned groups and important for statistical analysis

#Comment 5: In the method section, during stereotaxic surgery authors have administered ketamine 160 mg/kg to the mice. It seem very high dose to that animal which may cause the respiratory depression, hypothermia, muscular tremors, myoclonic jerking and cardiac arrest due to which mortality rate will increased. Literature survey suggests that the anesthetic dose of ketamine is ranges between 70-100 mg/kg, intraperitoneally. If there is any reference for administering this much amount of drug, please provide.

#Comment 6: The information that authors provided in the Introduction paragraph 5, have been reproduced as such in the Discussion paragraph 2. It's a big typographical error and need to be fixed.

#Comment 7: In results section, authors have mentioned about the chronic morphine treatment from day 2-6 in few groups and also chronic saline treatment in other groups. However, figures depict only chronic morphine and not the saline treatment (fig. 1a; fig. 2a; fig. 4d).

#Comment 8: The sentences in the manuscript are too long, confusing and incomplete. The overall english improvement is needed.

Minor comments:

#Comment 1: First line of abstract 'the lateral hypothalamus (LH) is physiologically critical in brain functions' has been written same in the introduction first paragraph.

#Comment 2: In abstract, the line "context-induced retrieval of withdrawal memory" has been repeated four times.

#Comment 3: In Introduction, the phrasal verb 'among them' has been repeatedly use by authors, please use alternative phrase to this.

#Comment 4: In Introduction section, VGLUT2 long form has not been mentioned when it has been used for first time, but it is mentioned for second time, kindly, change it.

#Comment 5: The abbreviated word NAc 'C' is capital in 6th paragraph of introduction.

#Comment 6: The abbreviation for nucleus accumbens (NAc) has been written twice in the 5th and 6th paragraphs of Introduction.

#Comment 7: In Immunohistochemistry and in situ hybridization, the word paraformaldehyde and its abbreviation PFA mentioned twice in 1st and 2nd paragraph.

#Comment 8: In result section, clozapine-n-oxide (CNO) abbreviation mentioned in result 1 that is ok. But it is also coming in result 2 description 2nd and 4th paragraph.

#Comment 9: The abbreviation for medium spiny projection neurons has been mentioned in 6th paragraph and it is also coming in discussion part. Authors have done same thing repeatedly. It is minor issues but its repetitions create more conflicts. Kindly, work on these minor typographical issues.

#Comment 10: In result description, the word fig. 1e is missing.

#Comment 11: Throughout the manuscript, authors have not written the long form of GABA. However, this word has been used repeatedly in the manuscript.

#Comment 12: The references have been cited in superscript numbering format, lots of reference has the spacing issues between last word and cited reference number.

#Comment 13: In method section of stereotaxic surgery, authors have not been provided the reference for stereotaxic coordinates at bregma for stereotaxic surgery. Please give the reference.

#Comment 14: The authors have not mentioned the meaning/long form of DREADD. However, this word has been used repeatedly in the manuscript.

Reviewer #3 (Remarks to the Author):

In this manuscript by Sheng, Lei and Chen et al., the authors show that glutamatergic neurons in LH are disinhibited by accumbens core D1 MSNs acting on LH GABA neurons to facilitate morphine withdrawal memory retrieval. To achieve this, the authors use a combination of IHC and FISH, optogenetics, chemogenetics, and patch-clamp recordings. Overall the authors present a complete suite of experiments that explains an important circuit in drug aversion. While I am generally supportive of the manuscript, I have two major concerns. First, I am not at all convinced this is “reward retrieval” and not “reward expression”. I think their data actually supports an expression interpretation. Second, the final experiment is lacking multiple important features, making it difficult to interpret. The foundation of the experiment is great, but the execution is poor enough that it makes the efforts moot.

Below I detail specific considerations.

Major considerations

1. The authors over emphasize the specificity of the D1 projection from core to LH. D2 neurons from shell do in fact project to LH, albeit less intensely than their D1 counterparts. Additionally, if the reasoning for investigating the D1-core populations was because of their projections to LH, then why not study the D1-shell population? I am not criticizing the choice of the core population (based on the literature writ large it seems reasonable), but the framing in the introduction does not well support their decision.
2. What was the rationale for the LH site selection for the experiments? The LH is a huge brain structure, and the site listed by the authors is quite rostral. In the introduction they indicate the potential importance of orexin signaling, but this far rostral site would largely be out of range for that neuron type. Like the first point, I am not questioning the site per se; NAc-core is known to project to those rostral regions, but they also project to mid and caudal zones too. So why rostral LH?
3. The authors claim to be testing memory retrieval, but to me, the test seems to be examining memory expression. If one wanted to look at retrieval, they would need to modulate neural activity during a retrieval event (like the post-test), then additionally test the animal the following day. If they disrupted retrieval, then the mice should show a strong CPA. If they did not affect retrieval, then they would show a weaker CPA. The interpretation would be that if retrieval is blocked, then the mice would not experience the first post-test as an extinction day. In contrast, if retrieval was not affected (and therefore the mice could appropriately update the associated memory), they should show a weaker CPA due to extinction.
4. The final experiment is very well crafted, but the results are difficult to interpret. Looking at the raw values, inhibition of LH GABA neurons did not seem to change baseline firing of LH glutamate neurons. Presumably we should have seen more APs in the inhibition alone, condition? This would then help explain why the additional excitation of the D1 core neurons had no effect; LH glutamate neurons were already firing more often. However, since the firing was lower than expected (close to the baseline reported in Fig 3), it is not very convincing. Secondly, I am surprised there was not a blue-only condition

as a positive control. Similarly, why was the baseline firing of the LH glutamate neurons not reported? It may be that this cohort of neurons had a slightly lower firing baseline, so maybe 5Hz is meaningful increase. But without a baseline or positive control, it is impossible to know.

Minor considerations

1. The authors demonstrate a very robust morphine CPA paradigm.
2. The CTB experiment was nicely done and compelling.
3. They authors use intersectional strategies very effectively and elegantly in the second half of the manuscript.

Reviewer #1

I have reviewed with eager interest the manuscript entitled “Disinhibited lateral hypothalamus glutamatergic neurons by circuits from nucleus accumbens contribute to withdrawal memory retrieval” by Huan Sheng et al. The manuscript shows that a specific population of LH glutamate neurons is FOC-activated during the retrieval of morphine withdrawal memory in mice previously conditioned to a context where they experienced negative affectivity of naloxone-precipitated withdrawal. Chemogenetic inhibition of glutamate neurons blocked the CPA. The neurons are indirectly activated by D1R-expressing GABAergic MSNs that project from the core region of the nucleus accumbens (NACC) to the LH. The MSNs are electrophysiologically coupled to LH GABA neurons, through which they activate LH glutamate neurons in a disinhibition mode. The whole functionality of the path was further confirmed by showing that blocking LH GABA neurons in the time of optogenetic stimulation of NACC MSNs abolished LH glutamate neuron responsivity. In my view, the methodological aspect of the work is remarkably rich and the results are outstanding. There are some comments, however, that may help improve the manuscript.

Comment:

As mentioned in the manuscript, there is evidence that LH glutamate neurons colocalize with orexin. The two neurotransmitters are not necessarily released always together as it has been shown that low-frequency photostimulation of orexin neurons caused release of glutamate, whereas high-frequency stimulation caused release of glutamate and orexin (Patricia Bonnavion, 2016). The question now is to what extent the involvement of LH glutamate neurons in naloxone-induced CPA is attributed to either glutamate or orexin neurotransmission? Although chemogenetic inhibition of glutamate neurons blocked CPA, this cannot dissociate glutamate effect from possible orexin effect. The Introduction provided some evidence that orexin may enhance glutamate neurotransmission, however, orexin has been implicated in the rewarding properties of drug.

Reply:

To what extent the involvement of LH glutamate neurons in naloxone-induced CPA is attributed to either glutamate or orexin neurotransmission is a very interesting question.

There are two evidences supporting that glutamate release from these LH neurons may make a major contribution to naloxone-induced CPA, whereas orexin released from these LH neurons may do less. The first evidence is from the distribution of glutamatergic and orexin neurons in the LH. The site we have studied in this work is LH rostral site where, based on reports, has a dense distribution of glutamatergic neurons with a sparse distribution of orexin neurons¹⁻⁴. The second evidence is from functional studies. Based on reports, although when receiving optical stimuli, LH glutamate neurons colocalized with orexin released both glutamate and orexin, the stimulation frequency required to trigger release of glutamate and orexin was different: orexin release only at upper frequencies, whereas glutamate release across all

frequencies⁵⁻⁷. This result suggests that if LH glutamatergic neurons receive excitatory input, they may always have a glutamate release, whereas orexin release from these neurons would depend on whether there is a higher presynaptic activity under CPA condition. Now, we have added a discussion about this point in the present revised version.

Comment:

Fig 1d. c-Fos immunoreactivity was observed only in 10% of LH glutamate neurons in naloxone-induced CPA group beyond those happened in control groups (~18% vs. ~8%). The same result came out of experiments related to Fig 3b. Although significant statistically, is it also significant biologically? Even if this biological effect was sought to be further confirmed by chemogenetic inhibition of glutamate neurons, this procedure probably dropped out virtually all glutamate neurons (see Fig 1f) not only the specific subset that was c-Fos activated during CPA.

Reply:

1. About “c-Fos immunoreactivity was observed only in 10% of LH glutamate neurons”, this situation was often observed under similar condition to our study. For example, Georgescu et al reported that morphine withdrawal increased c-Fos immunoreactivity in orexin cells about ~13% compared to control groups in the LH and this extent of increase had biological significance⁸.
2. As whether the influence of chemogenetic inhibition of glutamate neuron is due to the inhibition of all LH glutamate neurons, not only the specific subset that is c-Fos activated during CPA, we think that the influence of this chemogenetic inhibition of LH glutamate neurons is mainly due to the inhibition of the c-Fos activated LH glutamate neurons because in general, non-c-Fos immunostaining neurons are at resting state^{9,10}, meaning that they are not recruited during CPA, so even these resting neurons are inhibited by chemogenetic method, they should less contribute to the inhibition of CPA.

Comment:

LH contains both GABA interneurons and GABA projection neurons. The latter that projects to the VTA is rewarding when stimulated (Jennings, 2015). Therefore, its inhibition may have anti-reward effect. How can the authors exclude the possibility that D1-MSNs inhibit this population of LH GABA neurons (rather than GABA interneurons) and this way they cause CPA during morphine withdrawal?

Reply:

Indeed, as reported by Nieh et al, GABA projection neurons from the LH to the VTA is rewarding when being stimulated using photostimulation method¹¹. So when context activated D1-MSNs projecting to the LH, it was possible that D1-MSNs inhibited this population of LH GABA projection neurons and had anti-reward effect. However, Nieh et al also reported that in the real-time place preference/avoidance (RTPP/A) experiments, which was similar to our conditioned place aversion (CPA) experiments,

they did not detect any significant effects of the inhibition of this population of LH GABA projection neurons on behavior¹¹. This evidence suggests that context does not activate GABA projection neurons from the LH to the VTA, meaning that they are not recruited during CPA, so even D1-MSNs projecting to the LH have an inhibitory effect on GABA projection neurons from the LH to the VTA, this inhibition should contribute less to CPA.

Comment:

Previous indications regarding the involvement of LH glutamate neurons in aversive behavior may help justify the significance of studying their role also in withdrawal memory. For example, consider the following references: Jennings, 2013; Stamatakis 2016.

Reply:

Indeed, Jennings et al reported that photostimulation of LH glutamate neurons produced an aversion in the real-time place avoidance experiments¹². Optogenetic stimulation of LH-LHb glutamatergic fibers also produced an aversion in the real-time place avoidance experiments¹³. These results are consistent with our conditioned place aversion (CPA) experiments and further support the statement that the activation of LH glutamate neurons is aversive. Now, we have added a discussion in the present revised version.

Comment:

Fig 1g/line 167. Within-group comparisons are needed. Was CPA blocked COMPLETELY by chemogenetic inhibition of LH glutamate neurons?

Reply:

Now, we have made within-group comparison for Fig 1g. The result showed that after the inhibition of LH glutamate neurons by chemogenetic method, the average CPA score of the pre-test vs. the post-test had no statistical difference ($P=0.1768$), suggesting that CPA was blocked completely by chemogenetic inhibition of LH glutamate neurons. In addition, we also have made within-group comparison for other similar experiments.

Comment:

Fig 2d. c-Fos expression is not evident; especially in the Mor+Nal group it seems that expression is not that different from other control groups. Please consider replacing the representative figures or repeating the experiments or confirming the results by western blotting.

Reply:

Indeed, the representative figures of c-Fos expression in Fig 2d, especially in the Mor+Nal group, did not represent the average level of c-Fos expression in these groups. Now, we have replaced them in the present revised version.

Comment:

Fig 3d. More direct data needs to be provided showing that photostimulation of NACC MSNs activated the neurons. Therefore, the response of these neurons in the form of increased firing rates to optogenetic stimulation would be of interest. Alternatively, 2) experiments that show longer duration of photostimulation alone (without simultaneous electrical stimulation) is able to increase firing activity of LH glutamate neurons is required. Please consider providing a raster plot and PSTH for the response of LH glutamate neurons to MSN photostimulation. Providing data from a Light off + PTX condition would be of interest to see the effect of PTX alone. In addition, performing the same experiments in a non-labeled neuron, preferably glutamatergic one, would increase the reliability of the claim.

Reply:

Based on the suggestions by the reviewer, we added three experiments: (1) The influence of photostimulation of NAcC MSNs alone without simultaneous electrical stimulation on firing activity of LH glutamate neurons; (2) the effect of PTX with Light off on firing activity of LH glutamate neurons; (3) the effect of photostimulation of NAcC MSNs and the effect of PTX with Light off on action potential firing of LH non-labeled glutamatergic neurons. Now, we have added these results in the present revised version. In addition, we have provided a raster plot and PSTH for the response of LH glutamate neurons to photostimulation of NAcC MSNs in the present revised version.

Comment:

Fig 4b, 4C. 8% of glutamate and 85% of GABA neurons were inhibited by D1-MSNs photostimulation. 75% of glutamate neurons were inhibited by photostimulation of GABA neurons. Therefore, ~59% of glutamate neurons should be excited by activation of D1-MSNs. Considering that ~30% of D1-MSNs were Fos-activated by CPA (Fig 2e), it is expected that ~17.7% of LH glutamate neurons should be excited during CPA, which is very close to that found in experiments for Fig 1d (18%) or Fig 3b (15%). Does this estimation make sense?

Reply:

Yes, this estimation makes sense.

Comment:

Fig 5c. I suppose that the schematic of neuronal connections is not correct.

Reply:

So sorry for this mistake! Now, we have corrected it in the present revised version.

Comment:

Fig 5d. The data from light-off condition is required. However, comparing the baseline firing of LH glutamate neurons in Fig 3d (~4 AP/s) with that of Yellow-light-on condition in Fig 5d (~5 AP/s) shows no significant difference. Why did not LH GABA

photoinhibition cause any increase in the firing rates of glutamate neurons? Does it mean LH GABA neurons have no inhibitory effect on LH glutamate neurons in the resting state? If so, how come their inhibition by MSNs give rise to disinhibition of glutamate neurons?

Reply:

1. Yes, the data from light-off condition in Fig 5d as a control is required to show whether the inhibition of LH GABA neurons can induce an increase in firing of LH glutamate neurons, in order to reflect an inhibitory effect of LH GABA neurons on firing of LH glutamate neurons in the resting state. Now, we have added the data from light-off condition in the revised version.
2. The data from light-off condition in Fig 3d was from a population of cells, whereas the data from light-on in Fig 5d was from another population of cells, so we did not compare the difference between the data from light-off condition in Fig 3d and the data from light-on in Fig 5d. In the present revised version, we have made a comparison between the data from light-off condition and the data from light-on condition in the same population of cells.

Comment:

Although the manuscript was written in a sufficiently good way, it needs to be revised in terms of English writing.

Reply:

Now, we have revised the manuscript in terms of English writing.

Comment:

Consider rephrasing the title. The word “disinhibited” cannot (or at least seems odd to) come with “by”. What about using the singular form of “circuits”?

Reply:

Based on the suggestion by the reviewer, now we have rephrased the title as “A circuit from nucleus accumbens disinhibits lateral hypothalamus glutamatergic neurons contributing to morphine withdrawal memory expression”.

Comment:

Throughout the text, the word “conditioned” can be omitted form the expression “conditioned context-induced retrieval of withdrawal memory.” In addition, this expression was often repeated redundantly in the text (for an example see lines 210-214).

Reply:

In the present revised version, we have omitted “conditioned” and avoided repeating redundantly like lines 210-214.

Comment:

Line 203. Remove “we proposed.”

Reply:

In the present revised version, we have removed “we proposed”.

Comment:

Line 109. Swap the place of “projecting” and “neurons”.

Reply:

In the present revised version, we have swapped the place of “projecting” and “neurons” in line 109.

Comment:

Line 123. Remove “in morphine withdrawn mice”.

Reply:

In the present revised version, we have removed “in morphine withdrawn mice”

Comment:

Lines 125-129. Use “were trained to conditioned place aversion” or other expressions instead of “accepted the conditioned context training”.

Reply:

In the present revised version, we have used “were trained to CPA” instead of “accepted the conditioned context training” in Lines 125-129.

Comment:

Throughout the text, use “conditioning training” or other expressions instead of “conditioned context training”.

Reply:

In the present revised version, we have used “context training” instead of “conditioned context training” or removed it.

Comment:

Line 276. To study ...

Reply:

Sorry for this mistake. Now, we have corrected “To studied” with “To study” in the present revised version.

Comment:

Lines 292-296. The first (“to directly test...”) and last (“to examine...”) parts of the

sentence are almost the same, hence redundant.

Reply:

Now, we have rephrased it in the present revised version.

Comment:

Line 313. “and the activation of LH glutamatergic neurons are” can be removed with no harm to the sentence.

Reply:

Now, we have removed “and the activation of LH glutamatergic neurons are” in the present revised version.

Reviewer #2

This manuscript have been addressed an interesting hypothesis that the DA1-MSNs of nucleus accumbens core (NAcC) to lateral hypothalamus (LH) activate the LH glutamatergic neurons. The present study suggests that the activation of LH glutamatergic neurons plays an important role in the retrieval of morphine withdrawal memory. This activation of LH glutamatergic neurons is initiated by DA1-MSNs of NAcC via disinhibiting the local GABAergic neurons of LH. Methods were appropriately conducted. Saline and chronic morphine treatment were given to mice for 2- 6 days with increasing order of dose per day (10, 20, 30, 40, 40, 40 mg/kg, intraperitoneally). Outcome measures behavioral test of conditioned place aversion (CPA) score for the retrieval of withdrawal memory. Statistical results are fully presented in the text for the behavioral experiments. Data presentation is clear and convincing. Results show strong evidence for withdrawal memory retrieval in condition place aversion model, in which CPA score of morphine + naloxone group shows significantly higher than saline + saline, saline + naloxone and morphine + saline group. Appropriate references are cited.

Major Comments:

Comment 1:

In Introduction 6th paragraph, authors have mentioned that D2-MSNs in the shell of the NAc have no direct projections to the LH, references cited by authors does not support to this statement. However, this sentence is seems to be very contradictory to this line i.e. D2-MSNs in the shell of the NAc have direct projections to the LH (Please go through these references, Frazier et al., 2010; Li et al., 2018) and do corrections accordingly.

Reply:

1. Yes, the description “D2-MSNs in the shell of the NAc have no direct projections to the LH” in our manuscript is too absolute. In references cited by our manuscript, O'Connor et al. reported that in D2 receptor-eGFP animals, using CTB-593 retrograde tracing method to label NAc shell neurons projecting to the LH, they

found that in a total of 593 CTB-positive cells (i.e., LH-projecting cells) in the NAc shell, only 29 cells ($5.2\% \pm 1.1\%$) were colabeled with D2-eGFP¹⁴. Therefore, D2 neurons in the shell of the NAc have few direct projections to the LH. Now, we have revised it as “D2-MSNs in the shell of the NAc have few direct projections to the LH”.

2. We also read the references mentioned by the reviewer. The first paper by Frazier et al examined the Gene Expression Nervous System Atlas (GENSAT) image database of D1 and D2 neurons expressing green fluorescent protein from BAC transgenic mice and found that there was a dense distribution of the cell body of D2 neurons in NAc shell and a dense expression of D2 receptors in axonal terminals of the LH, so they speculated that D2 neurons appeared to send robust projections directly to the LH¹⁵. However, they did not have evidence supporting that these terminals were from NAc shell. Therefore, in this paper, they pointed out that detailed anatomical studies were needed to determine the output pathways of D2 MSNs from the NAc shell. In contrast, in the reference cited by us in the instruction, the authors performed a retrograde tracing study to labelled NAc neurons that project to the LH with injections of CTB-593 into the LH and demonstrated that D2-MSNs in the shell of the NAc had few direct projections to the LH¹⁴. As the second paper mentioned by the reviewer, this paper mainly involved the inputs to D2 MSNs of the NAc, rather than the outputs of D2 MSNs of the NAc¹⁶.

Comment 2:

In Introduction 2nd paragraph last line, authors are saying that the role of glutamatergic neurons of the LH in drug addiction remains unknown. However, the role of glutamatergic neurons in the LH have been contributed for the drug of abuse (Kindly, refer the article Chen et al., 2013). This is contrary to the statement provided by authors. Please go through the extensive literature survey and update the information.

Reply:

1. We have read the article by Chen et al¹⁷. They studied the role of glutamatergic input to the LH in ethanol intake, rather than the role of LH glutamatergic neurons. They found that when being injected into the LH, NMDA and AMPA both significantly increased ethanol intake and stimulated the expression of orexin in the LH. Now, we have cited this reference in the second paragraph involving the role of orexin in drug addiction in the revised version.
2. We again go through the extensive literature survey through PubMed and Google scholar. The role of glutamatergic neurons of the LH in drug addiction indeed is poorly documented. Now, in the present revised version, we use “the role of glutamatergic neurons of the LH in drug addiction remains poorly documented” to replace “the role of glutamatergic neurons of the LH in drug addiction remains unknown”.

Comment 3:

In Introduction section, there is no any reference in 4th paragraph. On what basis authors have provided the information, every information should have been included with appropriate citation of references.

Reply:

Now, we have added appropriate citation of references in the revised version.

Comment 4:

In the method section, authors have not provided the data regarding cannula verification, if authors have done cannula verification; please provide the original histological images and the sample size use in the experiment. It is very much necessary because after performing stereotaxic surgery by using coordinates from atlases reference to bregma, as for many instances cannula are not placed at right region of interest and leads to mislead of data collection. This also enables to decide the number of animals for the concerned groups and important for statistical analysis.

Reply:

In this study, we did not perform cannula implantation. We injected virus to brain regions (NAcC, LH) using stereotaxic surgery method. In this method, the skull was exposed with a small incision and holes were drilled to deliver virus with glass electrode. The coordinates of viral injection were determined according to the mouse brain atlas. After the injection, animals were sacrificed and slices containing the region of interest were prepared. The brain slices were imaged with a 10 X objective using a microscope to observe the site of virus fluorescent expression. If the expression of virus is not within the region of interest, the data of this animals was excluded from further analysis.

Comment 5:

In the method section, during stereotaxic surgery authors have administered ketamine 160 mg/kg to the mice. It seem very high dose to that animal which may cause the respiratory depression, hypothermia, muscular tremors, myoclonic jerking and cardiac arrest due to which mortality rate will increased. Literature survey suggests that the anesthetic dose of ketamine is ranges between 70-100 mg/kg, intraperitoneally. If there is any reference for administering this much amount of drug, please provide.

Reply:

We are so sorry for our typing mistake here. When we checked the dose of ketamine, we found that in fact, we used pentobarbital to anesthetize mice, rather than ketamine. One author of this manuscript just copied the sentence "Mice were anesthetized with ketamine and xylazine (160 mg/kg and 12 mg/kg body weight, respectively) before the stereotaxic surgery were performed" from our previous published paper¹⁸ to the manuscript. Now, we have corrected it in the present revised version.

Comment 6:

The information that authors provided in the Introduction paragraph 5, have been

reproduced as such in the Discussion paragraph 2. It's a big typographical error and need to be fixed.

Reply:

Now, we have removed the reproduced part of the Introduction paragraph 5 in the revised version.

Comment 7:

In results section, authors have mentioned about the chronic morphine treatment from day 2-6 in few groups and also chronic saline treatment in other groups. However, figures depict only chronic morphine and not the saline treatment (fig. 1a; fig. 2a; fig. 4d).

Reply:

Sorry, we did not describe it clearly. In these original figures, we labeled saline treatment in the bottom of the experimental timeline. Now, we have revised the upper text of the experimental timeline as "chronic morphine or saline treatment".

Comment 8:

The sentences in the manuscript are too long, confusing and incomplete. The overall English improvement is needed.

Reply:

Now, we have improved the manuscript in terms of English writing.

Minor comments:

#Comment 1: First line of abstract 'the lateral hypothalamus (LH) is physiologically critical in brain functions' has been written same in the introduction first paragraph.

Reply:

Now, we have rephrased it in the revised version.

#Comment 2: In abstract, the line "context-induced retrieval of withdrawal memory" has been repeated four times.

Reply:

Now, we have revised them to avoid these repeats in the revised version.

#Comment 3: In Introduction, the phrasal verb 'among them' has been repeatedly use by authors, please use alternative phrase to this.

Reply:

Now, we have rephrased it in the revised version.

#Comment 4: In Introduction section, VGLUT2 long form has not been mentioned when it has been used for first time, but it is mentioned for second time, kindly, change

it.

Reply:

Now, we have revised it.

#Comment 5: The abbreviated word NAc 'C' is capital in 6th paragraph of introduction.

Reply:

Now, we have fixed it in the revised version.

#Comment 6: The abbreviation for nucleus accumbens (NAc) has been written twice in the 5th and 6th paragraphs of Introduction.

Reply:

Now, we have revised it.

#Comment 7: In Immunohistochemistry and in situ hybridization, the word paraformaldehyde and its abbreviation PFA mentioned twice in 1st and 2nd paragraph.

Reply:

Now, we have revised it.

#Comment 8: In result section, clozapine-n-oxide (CNO) abbreviation mentioned in result 1 that is ok. But it is also coming in result 2 description 2nd and 4th paragraph.

Reply:

Now, we have revised it.

#Comment 9: The abbreviation for medium spiny projection neurons has been mentioned in 6th paragraph and it is also coming in discussion part. Authors have done same thing repeatedly. It is minor issues but its repetitions create more conflicts. Kindly, work on these minor typographical issues.

Reply:

Now, we have fixed it in the revised version.

#Comment 10: In result description, the word fig. 1e is missing.

Reply:

Now, we have fixed it in the revised version.

#Comment 11: Throughout the manuscript, authors have not written the long form of GABA. However, this word has been used repeatedly in the manuscript.

Reply:

Now, we have added the long form of GABA when it has been used for first time in the revised version.

#Comment 12: The references have been cited in superscript numbering format, lots of reference has the spacing issues between last word and cited reference number.

Reply:

Now, we have fixed it in the revised version.

#Comment 13: In method section of stereotaxic surgery, authors have not been provided the reference for stereotaxic coordinates at bregma for stereotaxic surgery. Please give the reference.

Reply:

Now, we have given the reference in the revised version.

#Comment 14: The authors have not mentioned the meaning/long form of DREADD. However, this word has been used repeatedly in the manuscript.

Reply:

Now, we have added the meaning/long form of DREADD when it has been used for first time in the revised version.

Reviewer #3

In this manuscript by Sheng, Lei and Chen et al., the authors show that glutamatergic neurons in LH are disinhibited by accumbens core D1 MSNs acting on LH GABA neurons to facilitate morphine withdrawal memory retrieval. To achieve this, the authors use a combination of IHC and FISH, optogenetics, chemogenetics, and patch-clamp recordings. Overall the authors present a complete suite of experiments that explains an important circuit in drug aversion. While I am generally supportive of the manuscript, I have two major concerns. First, I am not at all convinced this is “reward retrieval” and not “reward expression”. I think their data actually supports an expression interpretation. Second, the final experiment is lacking multiple important features, making it difficult to interpret. The foundation of the experiment is great, but the execution is poor enough that it makes the efforts moot. Below I detail specific considerations.

Comment:

The authors over emphasize the specificity of the D1 projection from core to LH. D2 neurons from shell do in fact project to LH, albeit less intensely than their D1 counterparts. Additionally, if the reasoning for investigating the D1-core populations was because of their projections to LH, then why not study the D1-shell population? I am not criticizing the choice of the core population (based on the literature writ large it seems reasonable), but the framing in the introduction does not well support their decision.

Reply:

1. Previous study showed that the activation of D2 neurons of NAc shell contributed

to context-induced retrieval of morphine withdrawal memory¹⁹. However, O'Connor EC et al reported that in a total of 593 CTB-positive cells (i.e., LH-projecting cells) in the NAc shell, only 29 cells (5.2% ± 1.1%) were co-labeled with D2-eGFP¹⁴, suggesting that D2 neurons in the shell of the NAc had few direct projections to the LH. Therefore, for the activation of LH glutamatergic neurons to participate in the expression of morphine withdrawal memory, the direct projection of D2-MSNs in the NAcSh to the LH may contribute less, so we have not studied whether a direct projection from D2 neurons of NAc shell to the LH participated in the activation LH glutamatergic neurons during the expression of drug withdrawal memory.

2. About “why not study the D1-shell population”, since previous study showed that the activation of D1-MSNs of NAc shell enhanced the rewarding effects of cocaine²⁰, whereas here, we focused on context-induced expression of morphine withdrawal memory, so we did not study the role of the D1-shell population in this work.
3. Now, we have revised the framing in the introduction to well support our decision.

Comment:

What was the rationale for the LH site selection for the experiments? The LH is a huge brain structure, and the site listed by the authors is quite rostral. In the introduction they indicate the potential importance of orexin signaling, but this far rostral site would largely be out of range for that neuron type. Like the first point, I am not questioning the site per se; NAc-core is known to project to those rostral regions, but they also project to mid and caudal zones too. So why rostral LH?

Reply:

1. In the introduction, we described the potential importance of orexin signaling just to introduce the research progress of the role of LH neurons in drug addiction. The purpose of this study is to examine the role of LH glutamatergic neurons in context-induced expression of morphine withdrawal memory, rather than the role of orexin neurons. Therefore, we did not select LH middle regions where there was a dense distribution of orexin neurons^{3,4}.
2. Yes, NAc-core is known to project to rostral, mid and caudal regions of the LH. The reason why we selected rostral LH was that in the CPA experiments, we examined the expression of c-Fos at each level of the LH and found that the increase in the expression of cFos was the most apparent in rostral LH. In addition, by injecting AAV-DIO-mGFP virus into NAc core, we found that rostral LH had the strongest projection terminals from NAc core. Therefore, we selected rostral LH.

Comment:

The authors claim to be testing memory retrieval, but to me, the test seems to examining memory expression. If one wanted to look at retrieval, they would need to modulate neural activity during a retrieval event (like the post-test), then additionally

test the animal the following day. If they disrupted retrieval, then the mice should show a strong CPA. If they did not affect retrieval, then they would show a weaker CPA. The interpretation would be that if retrieval is blocked, then the mice would not experience the first post-test as an extinction day. In contrast, if retrieval was not affected (and therefore the mice could appropriately update the associated memory), they should show a weaker CPA due to extinction.

Reply:

- 1) Here, using memory expression is also right. In conditioned place aversion (CPA) procedure, withdrawal-related memory consists of acquisition, retrieval /expression and reconsolidation or extinction phases. During acquisition, animals experience withdrawal symptoms in a chamber with distinctive context and then form associated memories of withdrawal symptoms and distinctive context. In the retrieval phase, context previously associated with withdrawal symptoms reactivates withdrawal memory and induces behavioral expression of the memory. Afterward, following the retrieval, withdrawal memory may experience two potentially dissociable but opposite processes: extinction that tends to weaken the original memory and reconsolidation which consolidates the original memory^{21,22}, depending on the duration of the exposure to context^{23,24}. Therefore, memory retrieval includes two stages: the reactivation of memory and the behavioral expression of memory retrieval. Therefore, in this memory paradigm, using both memory retrieval or memory expression is all right. Now, based on the comment by the reviewer, we have used memory expression to replace memory retrieval in the present revised version.
- 2) About the time of doing retrieval test, here we do it on the first day after training. This procedure examines the memory retrieval after memory formation, whereas doing it on the following day of the first retrieval test examines the memory retrieval after memory extinction or memory reconsolidation because after the first memory retrieval, the memory would experience extinction or reconsolidation^{21,22}.

Comment:

The final experiment is very well crafted, but the results are difficult to interpret. Looking at the raw values, inhibition of LH GABA neurons did not seem to change baseline firing of LH glutamate neurons. Presumably we should have seen more APs in the inhibition alone, condition? This would then help explain why the additional excitation of the D1 core neurons had no effect; LH glutamate neurons were already firing more often. However, since the firing was lower than expected (close to the baseline reported in Fig 3), it is not very convincing. Secondly, I am surprised there was not a blue-only condition as a positive control. Similarly, why was the baseline firing of the LH glutamate neurons not reported? It may be that this cohort of neurons had a slightly lower firing baseline, so maybe 5Hz is meaningful increase. But without a baseline or positive control, it is impossible to know.

Reply:

- 1) Indeed, as the reviewer pointed out, here, blue-only condition as a positive control was required, although we had this result in Figure 5e. Now, we have added a blue-only condition as a positive control in the present revised version. In addition, based on the suggestion by the reviewer, we examined the influence of the inhibition of LH GABA neurons on action potential firing of LH glutamate neurons in the resting state. The result shows that after the inhibition of LH GABA neurons, action potential firing of LH glutamate neurons increased. Now, we have added this result in the present revised version. This result combined with the result of the effect of GABA A receptor antagonist PTX on firing activity of LH glutamate neurons suggests that this cohort of neurons indeed had a lower firing baseline due to an inhibitory control from LH local GABAergic neurons on them.
- 2) About the firing in Fig 5d was lower than expected (close to the baseline reported in Fig 3), since the data from light-off condition in Fig 3d is from a population of cells, whereas the data from light-on in Fig 5d is from another population of cells, we did not compare the difference between the data from light-off condition in Fig 3d and the data from light-on in Fig 5d. In the present revised version, we have made a comparison between the data from light-off condition and the data from light-on condition in the same population of cells.

Comment:

Minor considerations

1. The authors demonstrate a very robust morphine CPA paradigm.
2. The CTB experiment was nicely done and compelling.
3. They authors use intersectional strategies very effectively and elegantly in the second half of the manuscript.

Reply:

Thanks for reviewer's comments.

References

- 1 Ziegler, D. R., Cullinan, W. E. & Herman, J. P. Distribution of vesicular glutamate transporter mRNA in rat hypothalamus. *J Comp Neurol* **448**, 217-229, doi:10.1002/cne.10257 (2002).
- 2 Negishi, K. *et al.* Distributions of hypothalamic neuron populations coexpressing tyrosine hydroxylase and the vesicular GABA transporter in the mouse. *J Comp Neurol* **528**, 1833-1855, doi:10.1002/cne.24857 (2020).
- 3 Wagner, D., Salin-Pascual, R., Greco, M. A. & Shiromani, P. J. Distribution of hypocretin-containing neurons in the lateral hypothalamus and C-fos-immunoreactive neurons in the VLPO. *Sleep Res Online* **3**, 35-42 (2000).
- 4 Peyron, C. *et al.* Neurons containing hypocretin (orexin) project to multiple neuronal systems. *J Neurosci* **18**, 9996-10015 (1998).
- 5 Bonnavion, P., Mickelsen, L. E., Fujita, A., de Lecea, L. & Jackson, A. C. Hubs and spokes of the lateral hypothalamus: cell types, circuits and behaviour. *The Journal of physiology* **594**, 6443-6462, doi:10.1113/jp271946 (2016).
- 6 Schone, C. *et al.* Optogenetic probing of fast glutamatergic transmission from hypocretin/orexin to histamine neurons in situ. *J Neurosci* **32**, 12437-12443, doi:10.1523/JNEUROSCI.0706-12.2012 (2012).
- 7 Schone, C., Apergis-Schoute, J., Sakurai, T., Adamantidis, A. & Burdakov, D. Coreleased orexin and glutamate evoke nonredundant spike outputs and computations in histamine neurons. *Cell Rep* **7**, 697-704, doi:10.1016/j.celrep.2014.03.055 (2014).
- 8 Georgescu, D. *et al.* Involvement of the lateral hypothalamic peptide orexin in morphine dependence and withdrawal. *The Journal of neuroscience : the official journal of the Society for Neuroscience* **23**, 3106-3111, doi:10.1523/jneurosci.23-08-03106.2003 (2003).
- 9 Cruz, F. C., Javier Rubio, F. & Hope, B. T. Using c-fos to study neuronal ensembles in corticostriatal circuitry of addiction. *Brain research* **1628**, 157-173, doi:10.1016/j.brainres.2014.11.005 (2015).
- 10 Hoffman, G. E., Smith, M. S. & Verbalis, J. G. c-Fos and related immediate early gene products as markers of activity in neuroendocrine systems. *Frontiers in neuroendocrinology* **14**, 173-213, doi:10.1006/frne.1993.1006 (1993).
- 11 Nieh, E. H. *et al.* Inhibitory Input from the Lateral Hypothalamus to the Ventral Tegmental Area Disinhibits Dopamine Neurons and Promotes Behavioral Activation. *Neuron* **90**, 1286-1298, doi:10.1016/j.neuron.2016.04.035 (2016).
- 12 Jennings, J. H., Rizzi, G., Stamatakis, A. M., Ung, R. L. & Stuber, G. D. The inhibitory circuit architecture of the lateral hypothalamus orchestrates feeding. *Science (New York, N.Y.)* **341**, 1517-1521, doi:10.1126/science.1241812 (2013).
- 13 Stamatakis, A. M. *et al.* Lateral Hypothalamic Area Glutamatergic Neurons and Their Projections to the Lateral Habenula Regulate Feeding and Reward. *J Neurosci* **36**, 302-311, doi:10.1523/JNEUROSCI.1202-15.2016 (2016).

- 14 O'Connor, E. C. *et al.* Accumbal D1R Neurons Projecting to Lateral Hypothalamus Authorize Feeding. *Neuron* **88**, 553-564, doi:10.1016/j.neuron.2015.09.038 (2015).
- 15 Frazier, C. R. & Mrejeru, A. Predicted effects of a pause in D1 and D2 medium spiny neurons during feeding. *The Journal of neuroscience : the official journal of the Society for Neuroscience* **30**, 9964-9966, doi:10.1523/jneurosci.2745-10.2010 (2010).
- 16 Li, Z. *et al.* Cell-Type-Specific Afferent Innervation of the Nucleus Accumbens Core and Shell. *Frontiers in neuroanatomy* **12**, 84, doi:10.3389/fnana.2018.00084 (2018).
- 17 Chen, Y. W., Barson, J. R., Chen, A., Hoebel, B. G. & Leibowitz, S. F. Glutamatergic input to the lateral hypothalamus stimulates ethanol intake: role of orexin and melanin-concentrating hormone. *Alcoholism, clinical and experimental research* **37**, 123-131, doi:10.1111/j.1530-0277.2012.01854.x (2013).
- 18 Yang, L. *et al.* Morphine selectively disinhibits glutamatergic input from mPFC onto dopamine neurons of VTA, inducing reward. *Neuropharmacology* **176**, 108217, doi:10.1016/j.neuropharm.2020.108217 (2020).
- 19 Zhu, Y., Wienecke, C. F., Nachtrab, G. & Chen, X. A thalamic input to the nucleus accumbens mediates opiate dependence. *Nature* **530**, 219-222, doi:10.1038/nature16954 (2016).
- 20 Lobo, M. K. *et al.* Cell type-specific loss of BDNF signaling mimics optogenetic control of cocaine reward. *Science* **330**, 385-390, doi:10.1126/science.1188472 (2010).
- 21 Rodriguez-Ortiz, C. J. & Bermudez-Rattoni, F. Determinants to trigger memory reconsolidation: The role of retrieval and updating information. *Neurobiol Learn Mem* **142**, 4-12, doi:10.1016/j.nlm.2016.12.005 (2017).
- 22 Abrari, K., Rashidy-Pour, A., Semnani, S. & Fathollahi, Y. Administration of corticosterone after memory reactivation disrupts subsequent retrieval of a contextual conditioned fear memory: dependence upon training intensity. *Neurobiol Learn Mem* **89**, 178-184, doi:10.1016/j.nlm.2007.07.005 (2008).
- 23 Pedreira, M. E. & Maldonado, H. Protein synthesis subserves reconsolidation or extinction depending on reminder duration. *Neuron* **38**, 863-869, doi:10.1016/s0896-6273(03)00352-0 (2003).
- 24 Suzuki, A. *et al.* Memory reconsolidation and extinction have distinct temporal and biochemical signatures. *J Neurosci* **24**, 4787-4795, doi:10.1523/JNEUROSCI.5491-03.2004 (2004).

REVIEWER COMMENTS

Reviewer #1 (Remarks to the Author):

I'd like to appreciate the authors providing the revised manuscript of their elegant work. It is much better now but still needs to be improved. My comments are as follows:

Line 561-566: If I am not mistaken, all morphine injections were given in the animals' homecages (or at least not in the CPP box) in a pretreatment mode. If so, this should be indicated in the Methods. Additionally, please indicate in Fig 1a and other timelines of the experiments which injections were given in the CPP maze.

Lines 576-581: I cannot understand why the mice were confined in the initially nonpreferred compartment after naloxone injection. Usually, the nonpreferred side is considered for drug-pairing if the drug is expected to be rewarding, and the preferred side is specified for drug-pairing if the drug is aversive. Additionally, it was stated that CPA score was calculated as time in the nonpreferred side minus time in the preferred side. If so, one can expect that all values in the pretest are already negative! Then, post-test CPA score would be still more negative in the group of mice conditioned with naloxone. Please make it more clear or consider reconstruction of the graphs.

Fig 2d: What percentage of the whole population of D1-expressing neurons in the NAc are FOS-activated during the expression of withdrawal memory (no matter if they labelled with CTB647 or not)? Is that significantly different from other groups?

Lines 267-271/Fig 3a, 3b: The methodology for this part should be mentioned in the text. As far as I understand, the protocol used here is the same as that used in Fig 1c, 1d in a group of mice prepared as in Fig 2g. If so, please mention this in the text or in the figure legend.

I suggest summarizing the main finding of the study in the first paragraph of the Discussion. Putting a graphical abstract/finding would also be of great interest.

The discussion generally remained silent about important issues such as the paired-pulse experiments results and the bulk of experiments done for demonstrating a disinhibition circuit between the D1-MSN input and the LH glutamate projection neurons. The results in Fig 4d-g and Fig 5d-e should be interpreted from the authors' point of view. PTX, yellow-light, blue-light, and their combinations have all the same effect on glutamate neurons (increased firing with equal effect size). How do these results indicate the presence of a disinhibition circuit? The physiological meaning of paired-pulse depression (increased release probability) following CPA memory expression with especial focus on possible underlying mechanisms should also be discussed.

Lines 346-347: The word "disinhibition" cannot be applied for GABA interneuron. Indeed, LH glutamate neurons were disinhibited, because disinhibition refers to the transient activation of a projection neuron

by reducing ongoing inhibition exerted by an interneuron. If you agree with this, I suggest reading over the text and correcting this term accordingly.

Line 464: "... under addiction condition, ..." Please use a more realistic word because addiction refers to a complex multi-aspect condition in humans. I suggest using "following morphine treatment" or "morphine dependence" or any other terms that show the experimental condition.

466-483: I generally agree with the authors' discussion of glutamate-orexin role in CPA, but have concern whether their latter reason applies to their study. How would the authors rule out the possibility that glutamate-orexin co-releasing neurons exhibit high-frequency firing under CPA memory expression? Although they have shown that during this condition the inhibitory drive on the glutamate neurons is removed, there may still be some excitatory drive from other sources on these neurons. C-FOS activation of LH glutamate neurons shown in Fig 1c may also indicate the presence of such high-frequency firing during the CPA memory expression. There also exists evidence that orexin is involved in fear response conditioning (<https://doi.org/10.1038/s41467-017-01782-z>) and aversive emotions (<https://doi.org/10.1186/s13041-021-00818-2>).

I think that for between- and within-group comparisons of CPA score the two-way repeated measures ANOVA is the right test (not an ordinary two-way ANOVA). If it was so, please indicate this in the statistical part of the Method. Please also indicate that the primary assumptions for using the parametric tests were passed by appropriate tests (like Shapiro-Wilk test).

Fig 5e: The graph consists two parts from different experiments each having its own control. So, instead of two-way ANOVA, independent statistical analyses should be performed for each part. Please use a paired t-test for the first part and a one-way repeated measures ANOVA for the second part. A line that separates the parts would help understanding why two control conditions were used.

Fig S1. Showing the involvement of D1-MSNs in CPA memory expression and lack of such an involvement for D2-MSNs is a principal result of this study. It would be better if it was moved to the main manuscript.

There are so many redundant details about the groups under comparison and their p values in figure legends. I suggest showing the differences using symbols in the graphs (as it is now) and only defining the symbols in the legends. The details in the Results section can be preserved. I also suggest using abbreviations for group names in the Results and Methods (for example: S = saline, M = morphine, N = naloxone).

The manuscript is generally well-written. The structure and the flow of information across the manuscript are all right. In spite of the great attempts made for improving the manuscript for English I think it still needs to be revised by a native English editor. In addition, it needs to become as succinct as possible with no harm to the informative nature of the text.

Reviewer #2 (Remarks to the Author):

The authors addressed most of the points previously raised. However, there are few critical points remaining.

Comment 1: Please state in manuscript, how many animals were excluded from the study due to improper injection of virus within the interested regions (NAcC and LH).

Comment 2: Include the route of administration for pentobarbital. Generally the anesthetic dose (mice) for pentobarbital is in the range of 40-50 mg/kg, ip, whereas in this study 100 mg/kg, ip dose is used. Please provide the appropriate references for specified dose.

Reviewer #3 (Remarks to the Author):

The authors have thoughtfully responded to my comments from the original review. I am particularly pleased of their shift to include "memory expression". I believe the manuscript has been substantially improved.

REVIEWER COMMENTS

Reviewer #1:

I'd like to appreciate the authors providing the revised manuscript of their elegant work. It is much better now but still needs to be improved. My comments are as follows:

Comment 1 :

Line 561-566: If I am not mistaken, all morphine injections were given in the animals' homecages (or at least not in the CPP box) in a pretreatment mode. If so, this should be indicated in the Methods. Additionally, please indicate in Fig 1a and other timelines of the experiments which injections were given in the CPP maze.

Reply:

Yes, all morphine or saline injections were given in the animals' home cage. We have added this description in the method section and have indicated it in Fig 1a and other timelines of the experiments where injections are given in the CPA maze in the revised version.

Comment 2 :

Lines 576-581: I cannot understand why the mice were confined in the initially nonpreferred compartment after naloxone injection. Usually, the nonpreferred side is considered for drug-pairing if the drug is expected to be rewarding, and the preferred side is specified for drug-pairing if the drug is aversive. Additionally, it was stated that CPA score was calculated as time in the nonpreferred side minus time in the preferred side. If so, one can expect that all values in the pretest are already negative! Then, post-test CPA score would be still more negative in the group of mice conditioned with naloxone. Please make it more clear or consider reconstruction of the graphs.

Reply:

Yes, as the reviewer points out, usually, the preferred side is specified for drug-pairing if the drug is aversive. In our study, considering the elimination of more influence of strong unconditioned preference on the results, we discarded mice with strong unconditioned aversion or preference for any compartment (i.e., > 80% of the session time) and kept mice with minor preference for one compartment. So, here, minor preferred side is specified for pairing context and aversive withdrawal.

Sorry, we did not describe it clearly. In fact, CPA score was defined as the time in the minor preferred compartment minus the time in the opposite compartment. So, all values in the pretest are positive and post-test CPA score were negative in the group of

mice conditioned with naloxone because the withdrawal conditioned training was performed in the minor preference compartment and the opposite compartment became more preferred compartment after withdrawal training.

Comment 3 :

Fig 2d: What percentage of the whole population of D1-expressing neurons in the NAc are FOS-activated during the expression of withdrawal memory (no matter if they labelled with CTB647 or not)? Is that significantly different from other groups?

Reply:

We calculated the percentage of the whole population of D1-expressing neurons in the NAc which were FOS-activated during the expression of withdrawal memory and it was significantly different from other groups. Now, we have added this result into the revised version.

Comment 4 :

Lines 267-271/Fig 3a, 3b: The methodology for this part should be mentioned in the text. As far as I understand, the protocol used here is the same as that used in Fig 1c, 1d in a group of mice prepared as in Fig 2g. If so, please mention this in the text or in the figure legend.

Reply:

Now, we have added the methodology in the text in the revised version.

Comment 5 :

I suggest summarizing the main finding of the study in the first paragraph of the Discussion. Putting a graphical abstract/finding would also be of great interest.

Reply:

Now, we have summarized the main finding of the study in the first paragraph of the Discussion in the revised version as follows:

“The main findings of the present study are that during context-induced expression of morphine withdrawal memory, LH glutamatergic neurons play an important role; NAc D1-MSNs projecting to the LH is an important upstream circuit of the activation of LH glutamatergic neurons; NAc D1-MSNs projecting to the LH activate LH glutamatergic neurons by removing the inhibitory effect of local GABAergic neurons on LH glutamatergic neurons. “

Comment 6 :

The discussion generally remained silent about important issues such as the paired-pulse experiments results and the bulk of experiments done for demonstrating a disinhibition circuit between the D1-MSN input and the LH glutamate projection neurons. The results in Fig 4d-g and Fig 5d-e should be interpreted from the authors' point of view. PTX, yellow-light, blue-light, and their combinations have all the same effect on glutamate neurons (increased firing with equal effect size). How do these results indicate the presence of a disinhibition circuit? The physiological meaning of paired-pulse depression (increased release probability) following CPA memory expression with especial focus on possible underlying mechanisms should also be discussed.

Reply:

About "How do these results indicate the presence of a disinhibition circuit", now, we have added a discussion in the revised version as follows:

"Our results showed that (1) single pulses of photostimulation (blue light) of NAcC D1-MSNs could elicit IPSCs in most LH GABAergic neurons and this IPSCs could be blocked by PTX, suggesting that NAcC D1-MSNs innervated LH GABAergic neurons; (2) single pulses of photostimulation (blue light) of LH local GABAergic neurons reliably elicited IPSCs in most LH glutamatergic neurons and this IPSCs could be blocked by PTX, suggesting that LH local GABAergic neurons innervated glutamatergic neurons in the LH; (3) single pulses of photoinhibition (yellow light) of LH local GABAergic neurons increased firing of action potentials, suggesting there was an inhibitory control of LH local GABAergic neurons on glutamatergic neurons of the LH; (4) after removing the inhibitory control of LH local GABAergic neurons on glutamatergic neurons of the LH using photoinhibition (yellow light), the increasing effect of photostimulation (blue light) of NAcC D1-MSNs on firing of LH glutamatergic neurons disappeared, suggesting that NAcC D1-MSNs projecting to the LH activated LH glutamatergic neurons via removing the inhibitory control of LH local GABAergic neurons on glutamatergic neurons of the LH. These results reveal the presence of a disinhibition circuit between NAcC D1-MSNs and LH glutamatergic neurons."

About possible mechanisms underlying increased GABA release from NAcC D1-MSNs projecting to the LH, now, we have added a discussion in the revised version as follows:

"In addition, our result showed that context could induce more GABA release from NAcC D1-MSNs terminals at presynaptic site of LH GABAergic neurons in morphine withdrawn mice, compared to normal mice. The mechanisms underlying this increase in GABA release from NAcC D1-MSNs projecting to the LH following CPA memory expression remains unknown. We speculate that it may be related to context-withdrawal association-induced strengthening of D1-MSNs projecting to the LH because in our previous study, we observed context-withdrawal association-induced strengthening in other projection

neurons⁶. ”

Comment 7 :

Lines 346-347: The word “disinhibition” cannot be applied for GABA interneuron. Indeed, LH glutamate neurons were disinhibited, because disinhibition refers to the transient activation of a projection neuron by reducing ongoing inhibition exerted by an interneuron. If you agree with this, I suggest reading over the text and correcting this term accordingly.

Reply:

Now, we have corrected “disinhibition” for GABA interneuron in the revised version.

Comment 8 :

Line 464: “... under addiction condition, ...” Please use a more realistic word because addiction refers to a complex multi-aspect condition in humans. I suggest using “following morphine treatment” or “morphine dependence” or any other terms that show the experimental condition.

Reply:

Now, we have used “morphine dependence” to replace “addiction condition” in the revised version.

Comment 9 :

466-483: I generally agree with the authors’ discussion of glutamate-orexin role in CPA, but have concern whether their latter reason applies to their study. How would the authors rule out the possibility that glutamate-orexin co-releasing neurons exhibit high-frequency firing under CPA memory expression? Although they have shown that during this condition the inhibitory drive on the glutamate neurons is removed, there may still be some excitatory drive from other sources on these neurons. C-FOS activation of LH glutamate neurons shown in Fig 1c may also indicate the presence of such high-frequency firing during the CPA memory expression. There also exists evidence that orexin is involved in fear response conditioning (<https://doi.org/10.1038/s41467-017-01782-z>) and aversive emotions (<https://doi.org/10.1186/s13041-021-00818-2>).

Reply:

We agree to the reviewer’s concern whether the second reason applies to our study. So we revised this part of discussion as follows:

“ There is evidence that a part of LH glutamatergic neurons co-release orexin and

glutamate from their axonal terminals (1,2,21). An interesting question is to what extent the involvement of LH glutamatergic neurons in context-induced expression of morphine withdrawal memory is attributed to either glutamate or orexin neurotransmission. The site we have studied in this work is LH rostral site where, based on reports, has a dense distribution of glutamatergic neurons with a sparse distribution of orexin neurons⁴⁹⁻⁵². This evidence appears to support that glutamate released from these LH neurons may make a major contribution to context-induced expression of morphine withdrawal memory, whereas orexin released from these LH neurons may do less.”

Comment 10 :

I think that for between- and within-group comparisons of CPA score the two-way repeated measures ANOVA is the right test (not an ordinary two-way ANOVA). If it was so, please indicate this in the statistical part of the Method. Please also indicate that the primary assumptions for using the parametric tests were passed by appropriate tests (like Shapiro-Wilk test).

Reply:

Now, we have indicated the two-way repeated measures ANOVA in the statistical part of the Method and also indicated that the primary assumptions for using the parametric tests passed by appropriate tests like Shapiro-Wilk test in the revised version.

Comment 11 :

Fig 5e: The graph consists two parts from different experiments each having its own control. So, instead of two-way ANOVA, independent statistical analyses should be performed for each part. Please use a paired t-test for the first part and a one-way repeated measures ANOVA for the second part. A line that separates the parts would help understanding why two control conditions were used.

Reply:

Now, we have used a paired t-test for the first part and a one-way repeated measures ANOVA for the second part, and a line was used to separates the parts in the revised version.

Comment 12 :

Fig S1. Showing the involvement of D1-MSNs in CPA memory expression and lack of such an involvement for D2-MSNs is a principal result of this study. It would be better if it was moved to the main manuscript.

Reply:

Now, we have moved the result of Fig S1 into the main manuscript in the revised version.

Comment 13 :

There are so many redundant details about the groups under comparison and their p values in figure legends. I suggest showing the differences using symbols in the graphs (as it is now) and only defining the symbols in the legends. The details in the Results section can be preserved. I also suggest using abbreviations for group names in the Results and Methods (for example: S = saline, M = morphine, N = naloxone).

Reply:

Now, we have only defined the symbols in the legends and used abbreviations for group names in the Results and Methods in the revised version.

Comment 13 :

The manuscript is generally well-written. The structure and the flow of information across the manuscript are all right. In spite of the great attempts made for improving the manuscript for English I think it still needs to be revised by a native English editor. In addition, it needs to become as succinct as possible with no harm to the informative nature of the text.

Reply:

Thanks for the suggestion! We also make attempts for improving the manuscript of English and becoming as succinct as possible with no harm to the informative nature of the text.

Reviewer #2:

The authors addressed most of the points previously raised. However, there are few critical points remaining.

Comment 1:

Please state in manuscript, how many animals were excluded from the study due to improper injection of virus within the interested regions (NAcC and LH).

Reply:

Now, we have stated the number of animals that are excluded from the study due to improper injection of virus within the interested regions in the manuscript. In addition, we provide a supplementary table showing the distribution of this number in different experiments.

Comment 2:

Include the route of administration for pentobarbital. Generally the anesthetic dose (mice) for pentobarbital is in the range of 40-50 mg/kg, ip, whereas in this study 100 mg/kg, ip dose is used. Please provide the appropriate references for specified dose.

Reply:

Now, we have included the route of administration for pentobarbital in the revised version.

Indeed, generally, the anesthetic dose (mice) for pentobarbital is in the range of 40-50 mg/kg, ip. So the original dose of pentobarbital sodium we applied for animal ethical approval was 40 mg/kg. However, during experiments, we found that this dose could well guarantee animals under anesthesia lasting 30 minutes required for single virus injection, but when two or four different viruses were injected, it usually needed anesthesia time for 60-100 minutes and this dose failed to meet the anesthesia need of this long-term virus injection. So based on literatures¹⁻⁴, we adjusted the dose of pentobarbital sodium from 40 mg/kg to 80-100 mg/kg and got animal ethical approval. This anesthesia does could guarantee animals under anesthesia lasting 100 minutes required for multiple virus injections without the death of animals, reducing the pain of animal during operation and keep normally postoperative recovery of anesthetized animals. Recently, we again checked the lethality of 100 mg/kg pentobarbital sodium and the duration of anesthesia. The results showed that no mice died after the application of this dose and the average duration of anesthesia of 80 mg/kg and 100 mg/kg of pentobarbital sodium was 73.14 ± 3.45 min and 108 ± 0.85 min, respectively. Now, we have provided references for this dose in the revised version.

Reviewer #3:**Comment:**

The authors have thoughtfully responded to my comments from the original review. I am particularly pleased of their shift to include "memory expression". I believe the manuscript has been substantially improved.

Reply:

Thanks for reviewer's comment.

References

- 1 Yang, F. *et al.* Activated astrocytes enhance the dopaminergic differentiation of stem cells and promote brain repair through bFGF. *Nat Commun* **5**, 5627, doi:10.1038/ncomms6627 (2014).
- 2 Yuan, Y. *et al.* Reward Inhibits Paraventricular CRH Neurons to Relieve Stress. *Curr Biol* **29**, 1243-1251 e1244, doi:10.1016/j.cub.2019.02.048 (2019).
- 3 Wang, J. *et al.* Postsynaptic RIM1 modulates synaptic function by facilitating membrane delivery of recycling NMDARs in hippocampal neurons. *Nature communications* **9**, 2267, doi:10.1038/s41467-018-04672-0 (2018).
- 4 Chen, X. J., Liu, Y. H., Xu, N. L. & Sun, Y. G. Multiplexed Representation of Itch and Mechanical and Thermal Sensation in the Primary Somatosensory Cortex. *J Neurosci* **41**, 10330-10340, doi:10.1523/JNEUROSCI.1445-21.2021 (2021).
- 5 Sun, J. *et al.* Excitatory SST neurons in the medial prelemniscal nucleus control repetitive self-grooming and encode reward. *Neuron* **110**, 3356-3373.e3358, doi:10.1016/j.neuron.2022.08.010 (2022).
- 6 Ma, Q. *et al.* A Conditioning-Strengthened Circuit From CA1 of Dorsal Hippocampus to Basolateral Amygdala Participates in Morphine-Withdrawal Memory Retrieval. *Frontiers in neuroscience* **14**, 646, doi:10.3389/fnins.2020.00646 (2020).

REVIEWERS' COMMENTS

Reviewer #1 (Remarks to the Author):

My concerns regarding the manuscript were addressed by the authors. I appreciate their patience in the review process and adding the requested data. Please, in Fig 2., label the figure to clearly specify that parts b and c are for D1-MSN and parts d and e are for D2-MSN.

REVIEWER COMMENTS

Reviewer #1:

My concerns regarding the manuscript were addressed by the authors. I appreciate their patience in the review process and adding the requested data. Please, in Fig 2., label the figure to clearly specify that parts b and c are for D1-MSN and parts d and e are for D2-MSN.

Reply:

Now, we have labeled Fig 2 to clearly specify that parts b and c are for D1-MSN and parts d and e are for D2-MSN in the revised version.